# Structure Matters: Dynamic Policy Gradient

**Sara Klein**[1,3]   **Xiangyuan Zhang**[2]   **Tamer Başar**[2]   **Simon Weissmann**[1]   **Leif Döring**[1]

[1]Institut for Mathematics, University of Mannheim, 68138 Mannheim
{simon.weissmann, leif.doering}@uni-mannheim.de

[2] Department of Electrical and Computer Engineering, and Coordinated Science Laboratory,
University of Illinois Urbana-Champaign, Urbana, IL 61801
{xz7, basar1}@illinois.edu

[3]August-Wilhelm Scheer Institute for Digital Products and Processes gGmbH, 66123 Saarbrücken
sara.klein@aws-institut.de

## Abstract

In this work, we study $\gamma$-discounted infinite-horizon tabular Markov decision processes (MDPs) and introduce a framework called dynamic policy gradient (DynPG). The framework directly integrates dynamic programming with (any) policy gradient method, explicitly leveraging the Markovian property of the environment. DynPG dynamically adjusts the problem horizon during training, decomposing the original infinite-horizon MDP into a sequence of contextual bandit problems. By iteratively solving these contextual bandits, DynPG converges to the stationary optimal policy of the infinite-horizon MDP. To demonstrate the power of DynPG, we establish its non-asymptotic global convergence rate under the tabular softmax parametrization, focusing on the dependencies on salient but essential parameters of the MDP. By combining classical arguments from dynamic programming with more recent convergence arguments of policy gradient schemes, we prove that softmax DynPG scales polynomially in the effective horizon $(1 - \gamma)^{-1}$. Our findings contrast recent exponential lower bound examples for vanilla policy gradient.

## 1   Introduction

The overarching goal in reinforcement learning (RL) is to train an agent that interacts with an unknown environment. This is mathematically modeled through a Markov decision process (MDP), where the agent actively learns a policy that, given a state, executes actions and receives rewards in return. The objective is to find a policy that maximizes the expected discounted reward. We categorize RL algorithms into two groups. Rather than the conventional separation between value-based and policy-based algorithms, we distinguish between algorithms that leverage the model's structure and those that do not. The first class utilizes the dynamic programming (DP) principle, such as value iteration, policy iteration, or Q-learning. Those algorithms optimize essentially single-period problems, where a single reward feedback is used in the update procedure and the future payoff is estimated by evaluation. The second class considers the entire multi-period problem at once and solves it using classical optimization, as in policy gradient (PG) methods [29, 27, 13, 10]. Over time, hybrid approaches integrating both methods, such as actor-critic (AC) methods, have consistently demonstrated superior performance in practical applications even though dynamic programming is used rather indirectly. In AC, the critic suggests a baseline, estimating either the value or action-value function. Dynamic programming becomes essential during the critic update process, effectively diminishing variance in gradient estimation. The essence of employing dynamic programming lies in the fact that parameter updates are not solely reliant on new trajectories, information is bootstrapped. Efforts of a broad research community have led to tremendous success of AC type algorithms (such as natural policy gradient (NPG), trust region policy optimization (TRPO), proximal policy optimization

39th Conference on Neural Information Processing Systems (NeurIPS 2025).

(PPO)) but these have not yet been backed by a sufficient theoretical understanding. This paper contributes to the basic theory of PG methods by studying the following questions:

*To what extent can the Markov property of an MDP be exploited directly to improve convergence of PG methods? How much improvement is gained compared to vanilla PG?*

This paper addresses both points by merging dynamic programming and policy gradient into dynamic policy gradient (DynPG) and analyzing its convergence behavior. DynPG can be seen as a hybrid RL algorithm that utilizes policy gradient to optimize a sequence of contextual bandits. In each iteration, the algorithm extends the horizon of the MDP by adding an additional epoch in the beginning and shifting trained policies to the future, as in applying the Bellman operator. A policy for the newly added epoch is trained by policy gradient where previous policies are used to determine future actions (cf., Figure 1). DynPG is not an AC method. It reduces the variance as far as possible by applying the previously trained, and therefore fixed, policies to generate a trajectory. This leads to stable estimates of the rewards-to-go and improves convergence behavior. In this work, we first present a general error analysis which is compatible with any optimization scheme to address each contextual bandit problem such as PG, NPG or policy mirror descent [30]. We then employ vanilla PG to present an explicit complexity bound under softmax parametrization.

## 1.1 Related work

The idea of using dynamic programming in searching the optimal policy is not new and dates back to the early 2000s, having been revisited several times in recent decades. Policy search by dynamic programming (PSDP) [3, 11, 21] is most closely related to DynPG and searches for optimal policies in a restricted policy class without explicitly using gradient ascent to find the optimal policy. Next to this CBMPI in [22] is also related to DynPG. Here the greedy policy is based on a previous value-iteration step. However in DynPG, no value iteration is necessary. There is a line of similar algorithms like approximate policy iteration (API), gradient temporal difference, policy dynamic programming and numerous other variations [5, 25, 26, 2, 9]. In these works, however, PG was not used as policy optimization step. In [12], a similar concept, also called dynamic policy gradient, was employed to tackle finite-time MDPs without discounting. Their motivation to utilize dynamic programming stemmed from seeking non-stationary optimal policies in finite-time MDPs. In contrast to their findings, our objective is to identify a stationary optimal policy and incorporate discounting into the framework. The present paper can be seen as a continuation of that line of research towards infinite-time horizon MDPs. Lastly, in their study of linear-quadratic continuous control and estimation problems, [33, 34, 35] proposed a receding-horizon policy gradient algorithm, which also integrates dynamic programming-based receding-horizon control with policy gradient methods.

The discount factor $\gamma \in (0,1)$ plays an essential role in the convergence behavior of RL algorithms, as they explicitly depend on the contraction property of the Bellman operator. The closer $\gamma$ to one, the slower the Bellman operator contracts. Similarly, the convergence of PG methods also heavily depends on $\gamma$ but establishing a clear dependence is generally challenging due to the non-convex optimization landscape. Early works only concern convergence to stationary points [18, 23, 17]. More recently, specific to tabular settings and the softmax parametrization, [1, 16] have derived upper bounds on the convergence rate to a global optimum by utilizing a gradient domination property [19]. As shown in [16], vanilla softmax PG has a sublinear convergence rate of $O(\epsilon^{-1})$ with respect to the error tolerance $\epsilon$. Dependencies on other salient but essential parameters of the MDP crucially affect the overall convergence behavior of PG methods, e.g., the effective horizon $(1 - \gamma)^{-1}$. Notably, [15] constructed a counterexample such that vanilla PG could take an exponential time (with respect to $(1 - \gamma)^{-1}$) to converge. Although the optimal solution can be reached in just $|\mathcal{S}|$ steps of exact value iteration in the counter example, vanilla softmax PG is very inefficient by not leveraging the inherent structure of the MDP. In contrast, for softmax DynPG, we establish an explicit dependency on the discount factor. The convergence rate under exact gradients scales with $(1 - \gamma)^{-4}$ up to logarithmic factors; thus, we delineate the unknown model-dependent constant $\mathcal{C}(\gamma)$ in the rate of vanilla softmax PG [16]. As a result, DynPG can efficiently address the counterexample of [15]; see Section 5. Table 1 summarizes the complexity bounds for softmax PG and softmax DynPG.

| algorithm | complexity bounds | reference |
|:---:|:---:|:---:|
| softmax PG upper bound | $O\big(\mathcal{C}(\gamma)(1-\gamma)^{-6}\epsilon^{-1}\big)$ | [16, Thm. 4] |
| softmax PG lower bound | $\lvert\mathcal{S}\rvert 2^{\Omega((1-\gamma)^{-1})}$ gradient steps for $\epsilon = 0.15$ | [15, Thm. 1] |
| softmax DynPG upper bound | $O\big((1-\gamma)^{-4}\epsilon^{-1}\log((1-\gamma)^{-2}\epsilon^{-1})\big)$ | [**our** Thm. 4.8] |

Table 1: Comparison of convergence rate under exact gradients

## 1.2 Main contributions and outline

The main contribution of this article is to introduce and analyze a way to directly combine policy gradient and dynamic programming ideas. The algorithm we introduce in Section 3 is called DynPG (dynamic policy gradient method). The algorithm (provably) circumvents recent worst case lower bound problems for standard PG that prove impracticability of plain vanilla PG in delicate environments. In Section 4 we provide a detailed error decomposition to show the convergence behavior of DynPG in general frameworks. For the tabular softmax parametrization we prove rigorous upper bounds on the convergence rate in Section 4.2 for the theoretical setting of exact gradients. Most importantly, the $\gamma$-dependence is harmless, DynPG does not suffer from exponential dependencies in $\gamma$! A simple example is presented in Section 3 to show in a sample based setting how DynPG can beat PG in environments that trick PG into so-called committal behavior.

## 2 Preliminaries

Let the tuple $\mathcal{M} = (\mathcal{S}, \mathcal{A}, p, r, \gamma)$ denote an MDP, where $\mathcal{S}$ represents the state space, $\mathcal{A}$ represents the action space, $p : \mathcal{S} \times \mathcal{A} \to \Delta(\mathcal{S})$ is the transition kernel mapping each state-action pair to the probability simplex of all possible next states $\Delta(\mathcal{S})$, $r : \mathcal{S} \times \mathcal{A} \to \mathbb{R}$ is the reward function mapping each state-action pair to a reward, $\mu \in \Delta(\mathcal{S})$ denotes the initial state distribution, and $\gamma \in (0, 1)$ is the discount factor. We assume that the reward is bounded such that $r(s, a) \in [-R^*, R^*]$ for all $(s, a) \in \mathcal{S} \times \mathcal{A}$ and the state and action spaces are finite. Hereafter, we use capital and lowercase letters to distinguish random state and actions from deterministic ones. For $x \in \mathbb{R}^d$, we denote its supremum norm by $\|x\|_\infty = \max_{i=1,\ldots,d} |x_i|$; we refer to Appendix A for complete notation conventions.

A policy $\pi : \mathcal{S} \to \Delta(\mathcal{A})$ is a mapping from a state $s \in \mathcal{S}$ to a distribution over the action space, i.e., $\pi(\cdot|s) \in \Delta(\mathcal{A})$. The set of all policies is denoted by $\Pi$. We refer to $h \in \mathbb{N}$ as the deterministic time-horizon, where the case $h = \infty$ corresponds to the standard infinite-horizon MDP. Non-stationary policies with time-horizon $h$ are denoted by $\boldsymbol{\pi}_h := \{\pi_{h-1}, \ldots, \pi_0\} \in \Pi^h$. Let $V_0 \equiv 0$ and, for all $h > 0$, define the $h$-step value function of policies $\boldsymbol{\pi}_h \in \Pi^h$ as

$$V_h^{\boldsymbol{\pi}_h}(\mu) = V_h^{\{\pi_{h-1},\ldots,\pi_0\}}(\mu) = \mathop{\mathbb{E}}_{\substack{S_0 \sim \mu, A_t \sim \pi_{h-t-1}(\cdot|S_t) \\ S_{t+1} \sim p(\cdot|S_t, A_t)}} \Big[ \sum_{t=0}^{h-1} \gamma^t r(S_t, A_t) \Big]. \tag{1}$$

When $\mu$ is a Dirac measure at $s$ we let $V_h^{\boldsymbol{\pi}_h}(s) := V_h^{\boldsymbol{\pi}_h}(\delta_s)$. Subsequently, we interpret a function $V : \mathcal{S} \to \mathbb{R}$ as a vector in $\mathbb{R}^{|\mathcal{S}|}$. Additionally, we use $V_h^\pi(\mu) := V_h^{\{\pi,\ldots,\pi\}}(\mu)$ to denote the value function of the stationary policy $\pi$ being applied $h$ times in a row. For $h = \infty$ the resulting infinite-horizon discounted MDP admits a stationary optimal policy [20]. We define $V_\infty^*(\mu) := \sup_{\pi \in \Pi} V_\infty^\pi(\mu)$ and use $\pi^*$ to denote a stationary policy that achieves $V_\infty^{\pi^*}(\mu) = V_\infty^*(\mu)$. In contrast, when $h$ is finite, the finite-horizon MDP optimization problem needs non-stationary optimal policies; thus, we define $V_h^*(\mu) := \sup_{\boldsymbol{\pi}_h \in \Pi^h} V_h^{\boldsymbol{\pi}_h}(\mu)$ and use $\boldsymbol{\pi}_h^* := \{\pi_{h-1}^*, \ldots, \pi_0^*\} \in \Pi^h$ to denote a sequence of policies that achieves $V_h^{\boldsymbol{\pi}_h^*}(\mu) = V_h^*(\mu)$.

For any function $V \in \mathbb{R}^{|\mathcal{S}|}$ and stationary policy $\pi$, the Bellman expectation operator $T^\pi : \mathbb{R}^{|\mathcal{S}|} \to \mathbb{R}^{|\mathcal{S}|}$ is defined for every $s \in \mathcal{S}$ by

$$T^\pi(V)(s) = \sum_{a \in \mathcal{A}} \pi(a|s) \Big( r(s, a) + \gamma \sum_{s' \in \mathcal{S}} p(s'|s, a) V(s') \Big).$$

The Bellman's optimality operator $T^* : \mathbb{R}^{|\mathcal{S}|} \to \mathbb{R}^{|\mathcal{S}|}$ is then for every $s \in \mathcal{S}$ given by

$$T^*(V)(s) = \max_{a \in \mathcal{A}} \left\{ r(s,a) + \gamma \sum_{s' \in \mathcal{S}} p(s'|s,a) V(s') \right\}.$$

The operators $T^\pi$ and $T^*$ are $\gamma$-contracting with unique fixed points denoted by $V_\infty^\pi$ and $V_\infty^*$, respectively. In addition, the optimal $h$-step value functions $V_h^*$ can be obtained by iteratively applying $T^*$ and, as $h \to \infty$, $V_h^*$ converges to $V_\infty^*$ (c.f. Lemma C.1).

**Vanilla Policy Gradient.** Policy gradient is a policy search method for which a family of differential parameterized policies $(\pi_\theta)_{\theta \in \mathbb{R}^d}$ is fixed, say $\pi_\theta(\cdot|s) := \pi(\cdot|s, \theta)$, and the optimal policy is searched using gradient ascent:

$$\theta_{n+1} = \theta_n + \eta \cdot \nabla_{\theta_n} V_\infty^{\pi_{\theta_n}}(\mu). \tag{2}$$

Here, $\eta > 0$ is the step size. Analyzing the convergence of vanilla PG is generally non-trivial due to the non-convexity of $V_\infty^{\pi_\theta}$ and the approximation error of the policy approximation family. Specialized to tabular MDPs, [1] has shown global convergence of vanilla PG under the softmax parametrization, and [16] established a non-asymptotic convergence rate. It should be emphasized that convergence is slow and constants can easily dominate convergence rates.

**Tabular Softmax Policy.** Policy gradient can be applied in the tabular setting (i.e. considering separately each state-action pair). Although the tabular setting is not used in practical applications, it is the most tractable setting for a complete mathematical analysis and sheds light on general principles. We introduce the logit function $\theta : \mathcal{S} \times \mathcal{A} \to \mathbb{R}$ and the softmax policy parametrized by $\theta \in \mathbb{R}^{|\mathcal{S}||\mathcal{A}|}$

$$\pi_\theta(a|s) = \frac{\exp(\theta(s,a))}{\sum_{a' \in \mathcal{A}} \exp(\theta(s,a'))}, \quad s \in \mathcal{S}, a \in \mathcal{A}. \tag{3}$$

The tabular softmax policy can approximate any stationary and therefore also any optimal policy, whereas other parameterizations such as neural networks may induce an approximation error. This error needs to be considered in the convergence analysis.

**Policy Search Framework.** Inspired by dynamic programming, the authors in [3, 11] proposed Policy Search by Dynamic Programming to search policies $\{\pi_{H-1}, \cdots, \pi_0\} \in \tilde{\Pi}^H$, where $H$ is the problem horizon given as an input to the algorithm and $\tilde{\Pi} \subseteq \Pi$ is the set of all deterministic policies. Formally, given $H$ and $\tilde{\Pi}$, PSDP computes

$$\tilde{\pi}_h^* = \operatorname{argmax}_{\pi_h \in \tilde{\Pi}} V_{h+1}^{\{\pi_h, \tilde{\boldsymbol{\pi}}_h^*\}}(\mu) = \operatorname{argmax}_{\pi_h \in \tilde{\Pi}} V_{h+1}^{\{\pi_h, \tilde{\pi}_{h-1}^*, \cdots, \tilde{\pi}_0^*\}}(\mu), \tag{4}$$

for $h = 0, \ldots, H - 1$. PSDP solves an $(h+1)$-step MDP initialized at an $S_0 \sim \mu$ in the iteration indexed by $h$. Specifically, it finds the optimal deterministic policy for selecting the first action $A_0$, denoted by $\tilde{\pi}_h^*$, but then all the remaining actions in the episode $\{A_1, \cdots, A_{h-1}\}$ are selected according to the sequence of deterministic policies $\tilde{\boldsymbol{\pi}}_h^* := \{\tilde{\pi}_{h-1}^*, \cdots, \tilde{\pi}_0^*\}$. Note that for all $h$, $\tilde{\pi}_h^*$ have been computed in previous iterations and are kept fixed. Compared to vanilla PG, PDSP exploits the Markovian property of the environment, rendering each iteration into solving a contextual bandit problem. While considering deterministic policies may suffice in tabular MDPs, no explicit computational procedures have been provided in [3, 11] to solve the optimization problem in (4). Further, determining the policy to be applied post-training is not immediately evident. The author in [21] proposse to apply the non-stationary policy $\tilde{\boldsymbol{\pi}}_H^*$ in a loop. Still, to solve the discussed infinite horizon MDP, a stationary policy is sufficient. We provide answers to these issues using DynPG.

# 3   Dynamic Policy Gradient

**The DynPG Algorithm.** DynPG starts by solving a one-step contextual bandit problem and then incrementally extends the problem horizon by one in each iteration, which is done by appending the new decision epoch in front of the current problem horizon (see the illustration in Figure 1). In each iteration, the parametrized (stochastic) policy responsible for sampling action $A_0$ from the newly-added epoch is optimized using gradient ascent. In subsequent time steps, DynPG applies the previous convergent policies to sample actions. Upon convergence of the current iteration, the

**Algorithm 1: DynPG**

**Input:** Initial state distribution $\mu$ and class of policies $(\pi_\theta)_{\theta \in \mathbb{R}^d}$.

**Result:** Approximation of $\pi^*$, denoted as $\hat{\pi}^*$.

1  Set $h \leftarrow 0$ and initialize $\Lambda \leftarrow []$;
2  **while** *Convergence criterion not met* **do**
3      Initialize $\theta_0$ (e.g., $\theta_0 \leftarrow 0$);
4      Design $\eta_h$ and $N_h$ (cf., Remark 4.4);
5      **for** $n = 0, \dots, N_h - 1$ **do**
6          Sample $G \approx \nabla_{\theta_n} V_{h+1}^{\{\pi_\theta, \Lambda\}}(\mu)$ (cf., Theorem C.3);
7          Update $\theta_{n+1} \leftarrow \theta_n + \eta_h G$;
8      **end**
9      Set $\hat{\pi}_h^* \leftarrow \pi_{\theta_{N_h}}$
10     Attach $\hat{\pi}_h^*$ at the beginning of $\Lambda$;
11     Set $h \leftarrow h + 1$;
12 **end**
13 Return $\hat{\pi}^* \leftarrow \hat{\pi}_{h-1}^*$ (the first element of $\Lambda$);

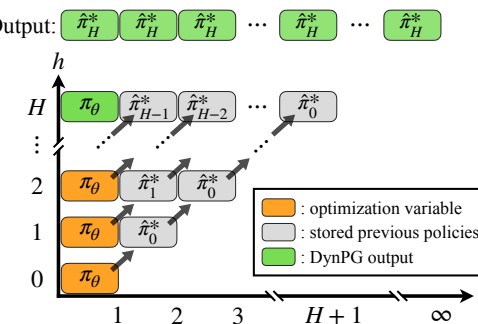

Figure 1: DynPG solves a sequence of contextual bandit problems, iteratively storing the convergent policies to memory and applying them accordingly as fixed policies in later iterations.

policy is stored in a designated memory location, which will be utilized in future iterations. We define $\hat{\boldsymbol{\pi}}_h^* := \{\hat{\pi}_{h-1}^*, \cdots, \hat{\pi}_0^*\} \in \Pi^h$ with convention $\hat{\pi}_0^* = \emptyset$ to denote the learned policies. DynPG returns the first policy in $\Lambda$ when certain user-defined convergence criteria have been met. For example, this is the case if the relative value improvements between two consecutive DynPG iterations are small i.e., $\|V_{h+1}^{\hat{\boldsymbol{\pi}}_{h+1}^*} - V_h^{\hat{\boldsymbol{\pi}}_h^*}\|_\infty \leq \epsilon$.

Hereafter, we let $V_0 \equiv 0$; the construction of DynPG implies

$$V_{h+1}^{\hat{\boldsymbol{\pi}}_{h+1}^*}(s) = T^{\hat{\pi}_h^*}(V_h^{\hat{\boldsymbol{\pi}}_h^*})(s) = \cdots = (T^{\hat{\pi}_h^*} \circ \cdots \circ T^{\hat{\pi}_0^*})(V_0)(s), \quad s \in \mathcal{S}, \tag{5}$$

which will be employed in the analyses of the convergence rate.

Compared to vanilla PG in (2), DynPG dynamically adjusts the episode horizon during the execution of the algorithm. This results in two notable advantages. Firstly, we observe that DynPG effectively reduces the variance in gradient estimation through the utilization of non-changing future policies. In contrast, the estimation of the gradient in vanilla PG involves assessing rewards-to-go based on the stationary policy $\pi_\theta$, which changes during training. Secondly, DynPG requires significantly fewer samples, a topic we discuss in detail later in this section. Moreover, each DynPG iteration has more benign optimization landscapes, as they are essentially contextual bandit problems. Specifically, DynPG has a simpler PG theorem (cf. Theorem C.3) and under softmax parametrization a smaller smoothness constant, enabling the selection of a more aggressive step size, $\eta_h$, to enhance convergence.

In contrast to PSDP, we specify the set of policies $\tilde{\Pi}$ to be a parameterized class of differentiable policies and a computational procedure, namely PG, for the inner loop. It is straight forward to incorporate function approximation (e.g. using neural networks) to the DynPG algorithm. This preserves the model-free optimization characteristic of gradient methods, while still exploiting the underlying structure of MDPs by Dynamic Programming. DynPG can be further modified with additional enhancements such as regularization, natural policy gradient or policy mirror descent in ever optimization epoch. In addition, we show that applying the policy of the last training epoch as stationary policy is sufficient. This is a non-trivial result and our analysis in Section 4 reveals that in general it requires additional training to obtain a good stationary policy compared to using the non-stationary policy $\hat{\boldsymbol{\pi}}_H^*$ in a loop as proposed in [21]. In the tabular softmax case we specify these additional computational cost explicitly.

**On the total sample complexity.** For each gradient step in standard policy gradient, one must run the MDP until termination or up to a stochastic horizon $H \sim \text{Geom}(1 - \gamma)$ to obtain an unbiased sample of the gradient [32]. DynPG, on the other hand, only requires $h$ interactions with the environment to sample an unbiased estimator of the gradient in epoch $h$. Thus, comparing other policy

gradient methods to DynPG solely based on the number of gradient steps is inadequate. Instead, for fairness, the number of samples (interactions with the environment) should be compared in practical implementations. For DynPG the total sample complexity is given by $\sum_{h=0}^{H-1}(h+1)N_h$. This results in a trade-off between increasing samples required for estimation and more accurate training to obtain convergence (cf. Section 4.1).

**A Numerical Example.** To demonstrate DynPG's effectiveness we present a numerical study of a canonical example where vanilla PG suffers from committal behavior. A detailed description of the MDP and the experimental setup can be found in Appendix B. We note that the theoretical analysis for softmax DynPG in Section 4.2 demonstrates its practical usefulness in fine-tuning the learning rates.

In the figure to the right we plotted the success probability to achieve an overall error in the value function of less than $\epsilon = 0.01$ from the optimal, i.e. $V_\infty^* - V_\infty^{\hat{\pi}_H^*} \leq 0.01$. The success probability is calculated by executing each algorithm 2000 times with randomly sampled initial states and the x-axis is the number of interactions with the environment used by both sample-based vanilla PG and sample-based DynPG. We observe that vanilla PG struggles solving this MDP, suffering from the high reward variance in certain states. Vanilla PG tends to concentrate on large rewards samples while DynPG circumvents this committal behavior by more accurate estimation of the future Q-values. It can be observed that the performance of DynPG and vanilla PG are similar for the first 400 interactions with the environment. However, vanilla

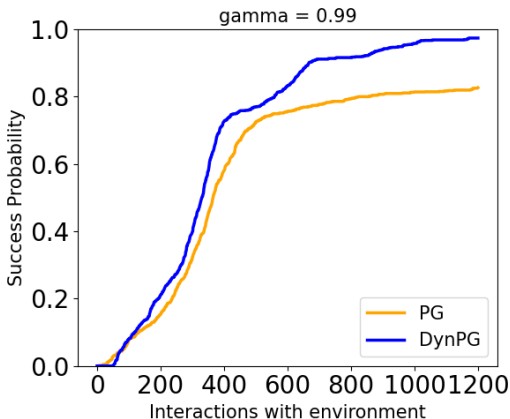

Figure 2: Success probability of achieving the sub-optimality gap of $\epsilon = 0.01$ in the overall error.

PG fails to converge to the optimal policy and can only reach the success probability of around $0.8$ under the given training budget. As usually for variants of vanilla PG, we expect that DynPG will not always be more efficient than vanilla PG. While vanilla PG is stuck in committal behavior in the environment chosen for the simulation, there are other (simpler) environments in which vanilla PG performs well. In that case our more complex algorithm will not beat vanilla PG or other equally well-performing variants, but we expect comparable performance instead.

**DynAC.** The DynPG algorithm requires to store a sequence of policies which for large problems might result in memory issues. To mitigate the problem there is a natural actor-critic type extension, that we call DynAC. Since an analysis is out of the scope of this article DynAC is described in the appendix, compare Section E.

## 4 Convergence Analysis

Our numerical example suggests improved convergence properties for DynPG. In this section we provide complexity bounds for exact gradients to keep notation simple. The reader with interest in bounds for estimated gradients is referred to Appendix F for the corresponding analysis.

In Section 4.1 four different layers of approximations were introduced that appear in the analysis of DynPG. Based on this we present the asymptotic global convergence of DynPG under the assumption of small enough optimization errors and rich enough parametrization class. In Section 4.2, we establish non-asymptotic global convergence rates of DynPG for the softmax parametrization under the assumption that exact gradients are available. We provide suitable choices of $H$, $N_h$ and $\eta_h$ such that DynPG achieves an error in the value functions of at most $\epsilon$. Proofs of all results are deferred to Appendix D.

## 4.1 Error Decomposition and General Convergence

The analysis of DynPG employs four different layers of approximations, including 1) adopting parametrizations incapable of modeling the optimal policy perfectly (approximation error), 2) truncating the infinite time horizon to a finite horizon (truncation error), 3) utilizing a finite number of gradient updates to approximately solve each optimization problem (accumulated optimization error), and 4) applying the first policy of $\hat{\pi}_h$ as a stationary policy in solving finite-horizon MDP (stationary policy error). We formally quantify these errors in the following proposition where the terms on the right-hand side of (6) correspond to the aforementioned errors, respectively.

**Proposition 4.1.** *The overall error of DynPG after $H$ iterations can be decomposed as follows*

$$
\begin{aligned}
\|V_\infty^* - V_\infty^{\hat{\pi}_H^*}\|_\infty &\leq \left\| V_\infty^* - \sup_\theta V_\infty^{\pi_\theta} \right\|_\infty + \left\| \sup_\theta V_\infty^{\pi_\theta} - \sup_{\theta_0,\ldots,\theta_{H-1}} V_H^{\{\pi_{\theta_{H-1}},\ldots\pi_{\theta_0}\}} \right\|_\infty \\
&\quad + \left\| \sup_{\theta_0,\ldots,\theta_{H-1}} V_H^{\{\pi_{\theta_{H-1}},\ldots\pi_{\theta_0}\}} - V_H^{\hat{\pi}_H^*} \right\|_\infty + \|V_H^{\hat{\pi}_H^*} - V_\infty^{\hat{\pi}_H^*}\|_\infty \\
&\leq \left\| V_\infty^* - \sup_\theta V_\infty^{\pi_\theta} \right\|_\infty + \frac{\gamma^H R^*}{1-\gamma} \\
&\quad + \sum_{h=0}^{H-1} \gamma^{H-h-1} \left\| \sup_\theta T^{\pi_\theta}(V_h^{\hat{\pi}_h^*}) - V_{h+1}^{\hat{\pi}_{h+1}^*} \right\|_\infty + \frac{1}{1-\gamma} \|V_{H+1}^{\hat{\pi}_{H+1}^*} - V_H^{\hat{\pi}_H^*}\|_\infty.
\end{aligned}
\tag{6}
$$

The error decomposition gives clear insights into algorithm design, for which it is necessary to discuss the practical implications of the four summands.

1. To control the approximation error a rich enough policy parametrization is required.

2. To keep the truncation error small, DynPG should run at least $H \approx \frac{\log(1-\gamma)}{\log(\gamma)}$ rounds.

3. The third summand shows that approximation errors in earlier iterations are discounted more than approximation errors in later iterations. To achieve optimal training efficiency, we will thus require a geometrically decreasing optimization error across the iterations of DynPG. Recall the sample complexity discussion at the end of Section 3 to note that a trade-off between more accurate training and increasing samples required for estimating the gradient must be made to obtain the best performance of DynPG.

4. DynPG approximates the value function by truncation at a fixed time $H$ and then replaces the optimal time-dependent policy by a stationary policy. As mentioned for PSDP, one could also apply the non-stationary policy $\hat{\pi}_h^*$ to approximate the value function. This would cause the fourth error term to vanish. Thus, in what follows, we distinguish between the **overall error** in (6) and the **value function error**:

$$
\begin{aligned}
\|V_\infty^* - V_H^{\hat{\pi}_H^*}\|_\infty &\leq \left\| V_\infty^* - \sup_{\theta \in \mathbb{R}^d} V_\infty^{\pi_\theta} \right\|_\infty + \frac{\gamma^H R^*}{1-\gamma} \\
&\quad + \sum_{h=0}^{H-1} \gamma^{H-h-1} \left\| \sup_{\theta \in \mathbb{R}^d} T^{\pi_\theta}(V_h^{\hat{\pi}_h^*}) - V_{h+1}^{\hat{\pi}_{h+1}^*} \right\|_\infty.
\end{aligned}
\tag{7}
$$

It is important to note that the factor $(1-\gamma)^{-1}$ in the stationary policy error causes an additional dependence on the effective horizon in the complexity bounds for softmax DynPG.

For the remainder of this section, we will proceed under the assumption of zero approximation error stemming from the parametrization. This condition generally holds true when the class of policies $(\pi_\theta)$ can effectively approximate all deterministic policies, such as achieved through tabular softmax. Under this circumstance, it follows that $T^* = \sup_\theta T^{\pi_\theta}$ and $\|V_\infty^* - \sup_\theta V_\infty^{\pi_\theta}\|_\infty = 0$. Subsequently, we demonstrate that a certain reduction in the optimization error is adequate for achieving global convergence within the parametrized policy space.

**Assumption 4.2.** *The class $(\pi_\theta)$ has zero approximation error and there exists a positive sequence $\epsilon_h$ such that*

- $\sum_{t=0}^{H-1} \gamma^{H-h-1} \epsilon_h \to 0$ *for* $H \to \infty$,

- *the policies obtained by DynPG satisfy* $\|T^*(V_h^{\hat{\boldsymbol{\pi}}_h^*}) - V_{h+1}^{\hat{\boldsymbol{\pi}}_{h+1}^*}\|_\infty \le \epsilon_h$ *for all* $h \ge 0$.

As an example for such a sequence one might think of $\epsilon_h = c\gamma^h$ for some $c > 0$. Under this assumption we prove convergence directly from the error decomposition in Proposition 4.1.

**Corollary 4.3.** *Let Assumption 4.2 hold. Then, Algorithm 1 generates a sequence of non-stationary policies* $\hat{\boldsymbol{\pi}}_H^* \in \Pi^H$ *that satisfy*

$$\|V_\infty^* - V_H^{\hat{\boldsymbol{\pi}}_H^*}\|_\infty \to 0 \quad \text{for } H \to \infty.$$

*Furthermore, the overall error vanishes in the limit:*

$$\|V_\infty^* - V_\infty^{\hat{\pi}_H^*}\|_\infty \to 0 \quad \text{for } H \to \infty.$$

**Remark 4.4.** *The condition* $\sum_{h=0}^{H-1} \gamma^{H-h-1} \epsilon_h \to 0$ *for* $H \to \infty$ *implies that the optimization error must decrease with increasing* $h$ *to guarantee convergence. Hence, training must become more precise over time. It is advisable to increase the number of gradient steps* $N_h$ *and decrease the step size* $\eta_h$.

## 4.2 Convergence Rates under Tabular Softmax Parametrization - Exact Gradients

In this section, we assume that all gradients are known explicitly, a strong assumption for practical use but standard for a rigorous analysis. The analysis is extended in Appendix F to estimated gradients.

One might ask whether Assumption 4.2 is reasonable and whether there are situations in which the condition on the algorithm holds. Indeed, we will show that this is the case for the softmax class of policies. More precisely, for any $\epsilon_h > 0$, we can specify a step size $\eta_h$ and a number of gradient steps $N_h$ such that Assumption 4.2 is satisfied. In a second step, we optimize $H$ and the error sequence $(\epsilon_h)$ to obtain the optimal sample complexity for DynPG under softmax parametrization, where we distinguish again between the value function error and the overall error.

**Assumption 4.5.** *Suppose that* $\mu(s) > 0$ *for all* $s \in \mathcal{S}$ *such that* $\left\|\frac{1}{\mu}\right\|_\infty < \infty$. *Furthermore, the parametrization in DynPG* $(\pi_\theta)_{\theta \in \mathbb{R}^{|\mathcal{S}||\mathcal{A}|}}$ *is chosen to be the tabular softmax parametrization introduced in Equation (3). We further assume a uniform distribution over the action space as the initialization of* $\theta_0$ *in Algorithm 1.*

**Theorem 4.6.** *Let Assumption 4.5 hold true and the gradient be accessed exactly. Let* $h \ge 0$, $\epsilon_h > 0$ *and* $\Lambda = \hat{\pi}_h^* \in \Pi^h$ *be a collection of* $h$ *arbitrary policies. Then, using step size* $\eta_h = \frac{1-\gamma}{2R^*(1-\gamma^{h+1})}$ *and gradient steps* $N_h = \frac{4R^*(1-\gamma^{h+1})|\mathcal{A}|^2}{(1-\gamma)\epsilon_h}\left\|\frac{1}{\mu}\right\|_\infty$ *in DynPG guarantees that the policy* $\hat{\pi}_h^*$ *of iteration* $h$ *achieves*

$$\|T^*(V_h^{\hat{\boldsymbol{\pi}}_h^*}) - V_{h+1}^{\hat{\boldsymbol{\pi}}_{h+1}^*}\|_\infty \le \epsilon_h.$$

By Corollary 4.3, we obtain from Theorem 4.6 convergence for softmax DynPG by choosing a sufficient decreasing error sequence $(\epsilon_h)$.

**Remark 4.7.** *The proof is adapted from [12, Lem. 3.4] to the discounted setting and is provided in Appendix D.2. Note that we will not directly apply [16, Thm. 4] with* $\gamma = 0$ *to prove this theorem for contextual bandits because the authors assumed rewards in* $[0, 1]$. *Even when we make the same assumption, the problem horizon increases with* $h$ *such that the contextual bandits we consider,* $V_{h+1}^{\{\pi_\theta, \Lambda\}}$, *will have maximal reward greater then 1 after the first iteration. Therefore we provide a more general framework with bounded rewards in* $[-R^*, R^*]$. *Furthermore, the smoothness constant used in [16] can be improved, which results in tighter bounds.*

In order to obtain complexity bounds from Theorem 4.6 for a given accuracy $\epsilon$ we optimize the total number of gradient steps $\sum_{h=0}^{H} N_h(\epsilon_h)$ with respect to $H$ and $(\epsilon_h)$ under the constraint that the overall error in (6) or the value function error in (7) is bounded by $\epsilon$. We provide a detailed solution to the described constrained optimization problem in Appendix D.3. Parts of the proof are motivated by a similar optimization problem solved in [28, Sec. 3.2]. We summarize the complexity bounds for both error types in the following.

**Theorem 4.8.** *[cf. Theorem D.6 and Theorem D.8 for detailed versions] Let Assumption 4.5 hold true and the gradient be accessed exactly. Choose $\epsilon > 0$.*

1. *Overall error: We can specify $H$, $(N_h)_{h=0}^H$ and $(\eta_h)_{h=0}^H$ such that,*

$$\sum_{h=0}^{H} N_h \leq \frac{48 R^* |\mathcal{A}|^2}{(1-\gamma)^3 \epsilon} \left\lceil \frac{\log(6R^*(1-\gamma)^{-2}\epsilon^{-1})}{\log(\gamma^{-1})} \right\rceil \left\| \frac{1}{\mu} \right\|_\infty$$

*accumulated gradient steps are required to achieve $\|V_\infty^* - V_\infty^{\hat{\pi}_H^*}\|_\infty \leq \epsilon$.*

2. *Value function error: We can specify $H$, $(N_h)_{h=0}^{H-1}$ and $(\eta_h)_{h=0}^{H-1}$ such that,*

$$\sum_{h=0}^{H-1} N_h \leq \frac{8 R^* |\mathcal{A}|^2}{(1-\gamma)^2 \epsilon} \left\lceil \frac{\log(2R^*(1-\gamma)^{-1}\epsilon^{-1})}{\log(\gamma^{-1})} \right\rceil \left\| \frac{1}{\mu} \right\|_\infty$$

*accumulated gradient steps are required to achieve $\|V_\infty^* - V_H^{\hat{\pi}_H^*}\|_\infty \leq \epsilon$.*

**Remark 4.9.** *We address each case individually and offer comprehensive versions of Theorem 4.8 in Theorem D.6 and Theorem D.8 delineating explicit selections for $H$, $(N_h)_{h=0}^H$, and $(\eta_h)_{h=0}^H$.*

To obtain the convergence behavior of DynPG in terms of $\gamma$ close to 1, in Lemma D.9 we derive that $\log(\gamma^{-1})$ is asymptotically equivalent to the term $(1-\gamma)$. Combined with Theorem 4.8, we find that the required gradient steps for $\gamma$ close to 1 behave like

$$O\big((1-\gamma)^{-4}\epsilon^{-1}\log((1-\gamma)^{-2}\epsilon^{-1})\big) \tag{8}$$

for the overall error and

$$O\big((1-\gamma)^{-3}\epsilon^{-1}\log((1-\gamma)^{-1}\epsilon^{-1})\big)$$

for the value function error. Compared to softmax PG, we observe an additional $\log(\epsilon^{-1})$ factor in the convergence rate and future research could explore the possibility of eliminating the log-factor. In terms of $\gamma$, however, DynPG offers a resilient upper bound, which, in comparison to vanilla PG, remains polynomial in the effective horizon.

# 5 Breaking the lower bound example with DynPG

In [15] a lower bound example was given for which softmax PG takes exponentially many steps in the expected horizon $(1-\gamma)^{-1}$ to converge. More precisely, it is shown that at least $|\mathcal{S}|2^{\Omega((1-\gamma)^{-1})}$ gradient steps are required to approximate the optimal value function with $\epsilon = 0.15$ accuracy. The constructed MDP is designed such that the unknown constant $C(\gamma)$ in the upper bound on softmax PG (see Table 1) is exponential in the effective time horizon $(1-\gamma)^{-1}$. To understand this behavior, we take a closer look at the definition of $C(\gamma) := \big(\min_{s\in\mathcal{S}} \inf_{n\geq 1} \pi^{\theta_n}(a^*(s)|s)\big)^{-1}$ in [16], where $a^*(s)$ denotes the optimal action in state $s$. To ensure small $C(\gamma)$ the probability of choosing the best action should not get close to 0 during training. But by construction in [15], finding the best action in state $s$ decreases as long as the probability of choosing the best action a previous state is not close enough to 1. This results in the phenomenon that the gradient steps required to converge towards $a^*(s)$ grows at least geometrically as $s$ increases. However, using DP one can solve the MDP within $|\mathcal{S}|$ steps of exact value iteration [15, Lemma 1]. This already implies that DynPG easily circumvents this exponential convergence time by employing DP. As future actions are determined by the previously trained policies, DynPG can evaluate the MDP under a non-stationary policy during training and thereby avoid the above described phenomenon. The upper bound on the complexity of softmax DynPG provides a theoretical proof that the needed gradient steps scale at most polynomial in $(1-\gamma)^{-1}$!

# 6 Conclusion and Future Work

This theory paper contributes to the understanding of PG methods by directly including dynamic programming to gradient based policy search. To this end, we have introduced DynPG, an algorithm

that directly combines DP and gradient ascent to solve $\gamma$-discounted infinite horizon MDPs. We provide mathematically rigorous performance estimates (for exact gradients in the main text, estimated gradients in Appendix F) in simplified situations which are typical for theory papers on PG methods. The main strength of the algorithm, in contrast to vanilla PG, is to provably avoid exponential $\gamma$-dependence. We have provided theoretical evidence that dynamic programming, combined with PG methods, can address well-known issues inherent to vanilla PG. This opens up a range of further avenues for future research, such as:

- Investigate whether the complexity bounds in the exact gradient setting are tight (numerically or theoretically) and explore the possibility of eliminating the log-factor.
- Analyze the variant DynAC from Section E. [14] present a sample complexity analysis of AC methods. Combining their results to control the critic error with the SGD analysis could result in convergence rates to achieve a small value function error in DynAC.
- Perform a simulation study of DynPG and DynAC on standard tabular environments and explore how numerical instability issues in standard PG can be overcome.
- Combine DynPG and DynAC with deep RL implementations for PG algorithms such as PPO, TRPO.

## Acknowledgements

The first author SK thankfully acknowledges the funding support by the Hanns-Seidel-Stiftung e.V. and is grateful to the DFG RTG1953 "Statistical Modeling of Complex Systems and Processes" for funding this research. The research of SK was also supported by the research project "SURF - Smart, user-centered regional flexibility platform for grid and market" funded by the Federal Ministry for Economic Affairs and Energy [03EI4092D]. The research of the second and third authors (XZ and TB) was sponsored in part by the US Army Research Office and was accomplished under Grant Number W911NF-24-1-0085.

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

# A Notations

| | |
|---|---|
| $\mathcal{S}$ | state space |
| $\mathcal{A}$ | action space |
| $p(\cdot\|s,a)$ | transition kernel in $(s,a)$; in $\Delta(\mathcal{S})$ |
| $r(s,a)$ | expected reward in $(s,a)$ |
| $\gamma \in (0,1)$ | discount factor |
| $R^* > 0$ | maximal absolute value of rewards |
| $\|x\|_\infty = \max_{i=1,\dots,d}\|x_i\|$ | supremum norm for $x \in \mathbb{R}^d$ |
| $\pi : \mathcal{S} \to \Delta(\mathcal{A}) \in \Pi$ | stationary policy (and stationary policy set) |
| $\boldsymbol{\pi}_h = \{\pi_{h-1},\dots,\pi_0\} \in \Pi^h$ | $h$-step non-stationary policy (and the set of all such policies) |
| $\mu$ | initial state distribution |
| $V_h^{\boldsymbol{\pi}_h}(\mu)$ | $h$-step discounted value function; policy $\boldsymbol{\pi}_h$, start distribution $\mu$ |
| $V_\infty^*(\mu)$ | optimal discounted value function |
| $\pi^*$ | optimal policy s.t. $V_\infty^*(\mu) = V_\infty^{\pi^*}(\mu)$ |
| $V_h^*(\mu)$ | optimal discounted $h$-step reward function |
| $\boldsymbol{\pi}_h^*$ | optimal $h$-step policy s.t. $V_h^*(\mu) = V_\infty^{\boldsymbol{\pi}_h^*}(\mu)$ |
| $V_0^\pi \equiv 0$ and $\boldsymbol{\pi}_0 = \{\}$ | conventions used in the manuscript |
| $Q_h^{\boldsymbol{\pi}_h}(s,a)$ | $h$-step Q-function |
| $T_\gamma^\pi$ | one-step Bellman operator under policy $\pi$ |
| $T_\gamma^*$ | one-step Bellman optimality operator |
| $\pi^V$ | stationary greedy policy after $V \in \mathbb{R}^{\|\mathcal{S}\|}$ |
| $Q^V(s,a)$ | Q-matrix obtained from $V \in \mathbb{R}^{\|\mathcal{S}\|}$ |
| $\eta_h, N_h$ | learning rate, number of training steps in epoch $h$ |
| $\Lambda$ | set of trained policies in the algorithm |
| $\hat{\pi}_h^*$ | trained policy by DynPG in epoch $h$ |
| $\boldsymbol{\hat{\pi}}_h^*$ | set of trained policy by DynPG from $h-1$ to $0$ |
| $\pi_\theta$ | stationary parametrized policy |
| $(\boldsymbol{\pi}_h)_\infty$ | apply $\boldsymbol{\pi}_h$ in a loop for the infinite horizon problem |

# B Details of the Numerical Example

The example is an extension of [24, Example 6.7], which has been used to compare different variants of Q-learning algorithms, as it suffers from overestimation of the Q-values. Since the overestimation problem in Q-learning and the committal behavior problem in policy gradient are closely related [8], we decided to use this example. The example is certainly artificial, as the number of actions and the rewards in the MDP are chosen to trap policy gradient from convergence. If we vary the MDP parameters from the current setting, DynPG consistently performs better, but the advantage over vanilla PG will become less significant. The code is available at `https://github.com/Sara-Klein/StructureMatters-DynPG`.

The MDP is defined as follows:

- The state space is given by $\mathcal{S} := \{0, \dots, 6\}$; States $0, 3, 6$ are the terminal states and states $1, 2, 4, 5$ are the initial states. We sample $s_0$ uniformly from $\{1, 2, 4, 5\}$.

- The action space is given by $\mathcal{A} := \{0, \dots, 299\}$.

- The state transitions and state-dependent actions are visualized in Figure 3. Each node represents a state, with squared ones being terminal states and elliptical ones being initial states. Each arrow represents an action that deterministically transits from one state to another. From state 1 to state 0, there is a total of 300 possible actions, succinctly visualized using the dots.

- Taking any $a \in \mathcal{A}$ from state 1 reaches state 0 and receives a reward of $r(1, a) \sim \mathcal{N}(-0.3, 10)$.

- Taking any of the 5 possible actions from state 4 reaches state 5 and receives the reward of $r(4, a) \sim \mathcal{N}(1.25, 1.25)$.

- Taking any of the 5 possible actions from state 5 reaches state 6 and receives the reward of $r(5, a) \sim \mathcal{N}(1.25, 1.25)$.

- Taking any of the 3 possible actions from state 2 receives the reward of $r(2, a) = 0$.

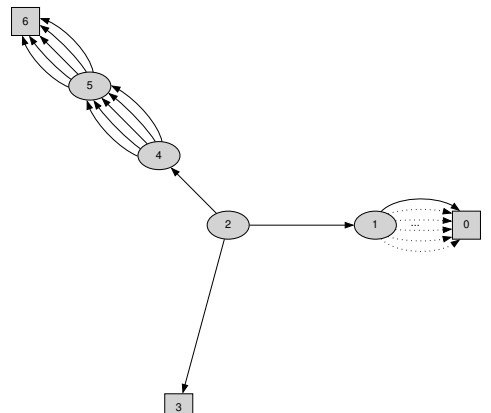

Figure 3: Visualization of the MDP state transitions.

**Experimental setup:** We evaluated the performance of (stochastic) vanilla PG and (stochastic) DynPG under two different discount factors, $\gamma = 0.9$ and $\gamma = 0.99$. We used the tabular softmax parametrization studied in the convergence analysis for both algorithms. In DynPG, we used the 1-batch Monte-Carlo estimator to sample the gradient according to Theorem A.3. In vanilla PG, we chose the classical REINFORCE 1-batch estimator with truncation horizon 3, such that the estimator is also unbiased due to the episodic setting (the maximum episode length in our example is 3).

In DynPG, we chose the step size $\eta_h$ and number of training steps $N_h$ according to Theorem D.6 and Theorem D.8, and only fine-tuned the constants 2 and 45:

$$\eta_h = 2\frac{1 - \gamma}{1 - \gamma^h}, \qquad N_h = \left\lceil 45\frac{1 - \gamma^{h+1}}{1 - \gamma} \right\rceil.$$

We want to emphasize that the choices of $\eta_h$ and $N_h$ consistently perform well under different choices of $\gamma$ (see therefore a second convergence picture with $\gamma = 0.9$ in Figure 4), which underscores that the algorithmic parameters developed in our theory also provide good guidance in practice. For a fair comparison, we fine-tuned $\eta = 2\frac{1-\gamma}{1-\gamma^6}$ for stochastic vanilla PG, which is much larger than the pessimistic $\eta = c * (1 - \gamma)^3$ suggested in [16].

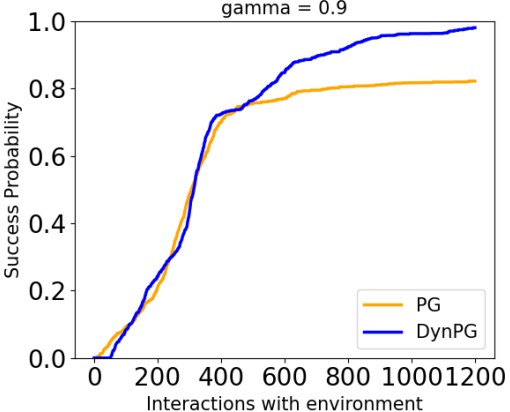

Figure 4: Success probability of achieving the sub-optimality gap of $\epsilon = 0.01$ in the overall error.

## C  Auxiliary Results

### C.1  Q-function, Advantage Function and Greedy Policy

First, we define the state-action value of $\boldsymbol{\pi}_{h-1} \in \Pi^{h-1}$ at any $(s,a) \in \mathcal{S} \times \mathcal{A}$ as

$$
\begin{aligned}
Q_h^{\boldsymbol{\pi}_{h-1}}(s,a) &:= \mathop{\mathbb{E}}_{\substack{S_{t+1} \sim p(\cdot|S_t, A_t) \\ A_t \sim \pi_{h-t-1}(\cdot|S_t)}} \left[ \sum_{t=0}^{h-1} \gamma^t r(S_t, A_t) \middle| S_0 = s, A_0 = a \right] \\
&= r(s,a) + \gamma \sum_{s' \in \mathcal{S}} p(s'|s,a) V_{h-1}^{\boldsymbol{\pi}_{h-1}}(s').
\end{aligned}
$$

For $h \geq 1$ and $\boldsymbol{\pi}_h \in \Pi^h$ we define the advantage function

$$
A_h^{\boldsymbol{\pi}_h}(s,a) := Q_h^{\boldsymbol{\pi}_{h-1}}(s,a) - V_h^{\boldsymbol{\pi}_h}(s) \tag{9}
$$

For $V \in \mathbb{R}^{|\mathcal{S}|}$, a greedy policy $\pi^V$ chooses the action that maximizes the $Q$-matrix[1]:

$$
Q^V(s,a) := r(s,a) + \gamma \sum_{s' \in \mathcal{S}} p(s'|s,a) V(s'), \quad s \in \mathcal{S}, \ a \in \mathcal{A}.
$$

By the definitions of the Bellman operators, it holds that a policy $\pi$ is greedy with respect to $V$ if and only if $T^\pi V = T^* V$ [6].

### C.2  Dynamic Programming - Finite Time Approximation

**Lemma C.1.** *Let $V_0 \equiv 0$. For any $h \geq 1$, it holds that*

(i) $V_h^*(s) = T^*(V_{h-1}^*)(s)$, *for all $s \in \mathcal{S}$,*

(ii) $\|V_\infty^* - V_h^*\|_\infty \leq \frac{\gamma^h}{1-\gamma} R^*$.

---

[1]When multiple actions are equally optimal, $\pi^V$ selects an arbitrary one among them.

*Proof.* C.f. [6, Prop. 1.2.1]

The first claim, follows directly from the definition of the Bellman optimality operator

$$V_h^*(s) = \sup_{\boldsymbol{\pi}_h \in \Pi^h} \sum_{a \in \mathcal{A}} \pi_0(a|s) \left( r(s,a) + \gamma \sum_{s' \in \mathcal{S}} p(s'|s,a) V_{h-1}^{\boldsymbol{\pi}_{h-1}}(s') \right)$$

$$= \max_{a \in \mathcal{A}} \left( r(s,a) + \gamma \sum_{s' \in \mathcal{S}} p(s'|s,a) \sup_{\boldsymbol{\pi}_{h-1} \in \Pi^{h-1}} V_{h-1}^{\boldsymbol{\pi}_{h-1}}(s') \right)$$

$$= T^*(V_{h-1}^*)(s).$$

For the second part, we fix $h \geq 0$. Recall that $\pi^* \in \Pi$ denotes a stationary optimal policy for the infinite-time problem and $\boldsymbol{\pi}_h^* \in \Pi_h$ an optimal non-stationary policy.

We divide the proof of the second claim into two cases.

Case 1: Assume that $V_\infty^*(s) - V_h^*(s) \leq 0$ for all $s \in \mathcal{S}$. Note that $V_\infty^*(s) \geq V_\infty^{(\boldsymbol{\pi}_h^*)_\infty}(s)$, where $(\boldsymbol{\pi}_h^*)_\infty$ denotes that we apply the finite time policy $\boldsymbol{\pi}_h^*$ in a loop for the infinite time problem. We have

$$V_h^{\boldsymbol{\pi}_h^*}(s) - V_\infty^*(s) \leq V_h^{\boldsymbol{\pi}_h^*}(s) - V_\infty^{(\boldsymbol{\pi}_h^*)_\infty}(s)$$

$$= - \underset{\substack{S_0=s,A_t \sim \pi_{h-(t \bmod h)-1}(\cdot|S_t) \\ S_{t+1} \sim p(\cdot|S_t,A_t)}}{\mathbb{E}} \left[ \sum_{t=h}^{\infty} \gamma^t r(S_t, A_t) \right]$$

$$\leq \sum_{t=h}^{\infty} \gamma^t R^* = \frac{\gamma^h}{1-\gamma} R^*,$$

where $R^*$ bounds the absolute value of rewards.

Case 2: Assume that $V^*(s) - V_h^*(s) > 0$ for all $s \in \mathcal{S}$. Note that $V_h^{\boldsymbol{\pi}_h^*}(s) \geq V_h^{\pi^*}(s)$ due to the optimality of $\boldsymbol{\pi}_h^*$ over the finite-time horizon. Further, by the definition of $V_\infty^*(s) = V_\infty^{\pi^*}(s)$, we have

$$V_\infty^*(s) - V_h^{\boldsymbol{\pi}_h^*}(s) \leq V_\infty^{\pi^*}(s) - V_h^{\pi^*}(s)$$

$$= \underset{\substack{S_0=s,A_t \sim \pi^*(\cdot|S_t) \\ S_{t+1} \sim p(\cdot|S_t,A_t)}}{\mathbb{E}} \left[ \sum_{t=h}^{\infty} \gamma^t r(S_t, A_t) \right]$$

$$\leq \sum_{t=h}^{\infty} \gamma^t R^* = \frac{\gamma^h}{1-\gamma} R^*.$$

Hence, we arrive at

$$\|V_\infty^* - V_h^{\boldsymbol{\pi}_h^*}\|_\infty = \max_{s \in \mathcal{S}} |V_\infty^*(s) - V_h^{\boldsymbol{\pi}_h^*}(s)| \leq \frac{\gamma^h}{1-\gamma} R^*.$$

$\square$

### C.3 Error bound for stationary policy

The following Lemma is inspired by error bounds presented in [6, Sec. 1.3]. The purpose of this result is to establish validation of applying the last policy trained in DynPG as stationary policy.

**Lemma C.2.** *For any $V \in \mathbb{R}^{|\mathcal{S}|}$ and policy $\pi \in \Pi$ it holds that*

$$\|V - V_\infty^\pi\|_\infty \leq \frac{\|T^\pi(V) - V\|_\infty}{1-\gamma}.$$

*Proof.* Consider any state $s \in \mathcal{S}$. Since $V_\infty^\pi$ is the unique fixed point of the operator $T^\pi$, we have

$$V_\infty^\pi(s) = T^\pi(V_\infty^\pi)(s) = \sum_{a \in \mathcal{A}} \pi(a|s) r(s,a) + \gamma \sum_{a \in \mathcal{A}} \pi(a|s) \sum_{s' \in \mathcal{S}} p(s'|s,a) V_\infty^\pi(s').$$

This implies that

$$
\begin{aligned}
& V_\infty^\pi(s) - V(s) \\
&= \sum_{a\in\mathcal{A}} \pi(a|s)r(s,a) + \gamma \sum_{a\in\mathcal{A}} \pi(a|s) \sum_{s'\in\mathcal{S}} p(s'|s,a)V_\infty^\pi(s') - V(s) \\
&= \sum_{a\in\mathcal{A}} \pi(a|s)r(s,a) + \gamma \sum_{a\in\mathcal{A}} \pi(a|s) \sum_{s'\in\mathcal{S}} p(s'|s,a)(V_\infty^\pi(s') - V(s') + V(s')) - V(s) \\
&= \sum_{a\in\mathcal{A}} \pi(a|s)r(s,a) + \gamma \sum_{a\in\mathcal{A}} \pi(a|s) \sum_{s'\in\mathcal{S}} p(s'|s,a)V(s') \\
&\quad + \gamma \sum_{a\in\mathcal{A}} \pi(a|s) \sum_{s'\in\mathcal{S}} p(s'|s,a)(V_\infty^\pi(s') - V(s')) - V(s) \\
&= T^\pi(V)(s) - V(s) + \gamma \sum_{a\in\mathcal{A}} \pi(a|s) \sum_{s'\in\mathcal{S}} p(s'|s,a)(V_\infty^\pi(s') - V(s')),
\end{aligned}
$$

for any $s \in \mathcal{S}$. We define the mappings $s \mapsto g_\pi(s) = T^\pi(V)(s) - V(s)$ and $s \mapsto J_\pi(s) = V_\infty^\pi(s) - V(s)$, $s \in \mathcal{S}$. Then, the above equation simplifies to

$$
J_\pi(s) = g_\pi(s) + \gamma \sum_{a\in\mathcal{A}}\sum_{s'\in\mathcal{S}} \pi(a|s)p(s'|s,a)J_\pi(s').
$$

By definition $J_\pi$ satisfies

$$
J_\pi(s) = \sum_{a\in\mathcal{A}} \pi(a|s)\left(g_\pi(s) + \gamma \sum_{s'\in\mathcal{S}} p(s'|s,a)J_\pi(s')\right)
$$

for all $s \in \mathcal{S}$, and therefore is a solution of the Bellman equation with an auxiliary reward function $(s,a) \mapsto \tilde{r}(s,a) = g_\pi(s)$. Note that this reward function is also bounded and by the uniqueness of the solution of the Bellman equation it has to hold that

$$
J_\pi(s) = \underset{\substack{S_0=s, A_t\sim\pi(\cdot|S_t)\\ S_{t+1}\sim p(\cdot|S_t,A_t)}}{\mathbb{E}}\left[\sum_{k=0}^\infty \gamma^k g_\pi(S_k)\right] = g_\pi(s) + \sum_{k=1}^\infty \gamma^k \underset{\substack{S_0=s, A_t\sim\pi(\cdot|S_t)\\ S_{t+1}\sim p(\cdot|S_t,A_t)}}{\mathbb{E}}\left[g_\pi(S_k)\right].
$$

Define $\underline{\beta} = \min_{s\in\mathcal{S}} g_\pi(s)$ and $\overline{\beta} = \max_{s\in\mathcal{S}} g_\pi(s)$. Then,

$$
\frac{\underline{\beta}}{1-\gamma} \le g_\pi(s) + \frac{\gamma\underline{\beta}}{1-\gamma} \le J_\pi(s) \le g_\pi(s) + \frac{\gamma\overline{\beta}}{1-\gamma} \le \frac{\overline{\beta}}{1-\gamma}.
$$

It follows that for any $s \in \mathcal{S}$,

$$
|J_\pi(s)| \le \frac{\max_{s\in\mathcal{S}}|g_\pi(s)|}{1-\gamma}.
$$

By definition of $g_\pi$ and $J_\pi$ this yields the claim. $\qquad\square$

### C.4 Policy Gradient Theorem for DynPG

**Theorem C.3.** *Suppose that $\pi_h \in \Pi^h$ is fixed for any $h \ge 0$, with the convention $\pi_0 = \emptyset$. Then, for any differentiable parametrized family, say $(\pi_\theta)_{\theta\in\mathbb{R}^d}$ (i.e. $\theta \mapsto \pi_\theta(s,a)$, $\theta \in \mathbb{R}^d$ is differentiable for all $(s,a) \in \mathcal{S}\times\mathcal{A}$), it holds that*

$$
\nabla_\theta V_{h+1}^{\{\pi_\theta, \boldsymbol{\pi}_h\}}(s) = \underset{S=s, A\sim\pi_\theta(\cdot|S)}{\mathbb{E}}\left[\nabla_\theta \log(\pi_\theta(A|S))Q_{h+1}^{\boldsymbol{\pi}_h}(S,A)\right].
$$

*Proof.* The proof can be found in [12, Thm A.6]. However, we will need an index shift and additionally to consider the discount factor $\gamma$. An $h$-step trajectory $\tau_h = (s_0, a_0, \ldots, s_{h-1}, a_{h-1})$ under policy $\{\pi_\theta, \boldsymbol{\pi}_h\}$ and initial state distribution $\delta_s$ occurs with probability

$$
p_s^{\{\pi_\theta, \boldsymbol{\pi}_h\}}(\tau_h) = \delta_s(s_0)\pi_\theta(a_0|s_0)\prod_{k=1}^{h-1} p(s_k|s_{k-1}, a_{k-1})\pi_k(a_k|s_k).
$$

Then, the log trick yields that

$$\nabla_\theta \log(p_s^{\{\pi_\theta, \boldsymbol{\pi}_h\}}(\tau_h))$$

$$= \nabla_\theta \Big( \log(\delta_s(s_0)) + \log(\pi_\theta(a_0|s_0)) + \sum_{k=1}^{h-1} \log(p(s_k|s_{k-1}, a_{k-1})) + \log(\pi_k(a_k|s_k)) \Big)$$

$$= \nabla_\theta \log(\pi_\theta(a_0|s_0)).$$

Let $\mathcal{W}_h$ be the set of all trajectories from $0$ to $h-1$. Then, the set $\mathcal{W}_h$ is finite due to the assumption that state and action space are finite. For $s \in \mathcal{S}$ we have

$$\nabla_\theta V_{h+1}^{\{\pi_\theta, \boldsymbol{\pi}_h\}}(s)$$

$$= \nabla_\theta \sum_{\tau_h \in \mathcal{W}_h} p_s^{\{\pi_\theta, \boldsymbol{\pi}_h\}}(\tau_h) \sum_{k=0}^{h-1} \gamma^k r(s_k, a_k)$$

$$= \sum_{\tau_h \in \mathcal{W}_h} p_s^{\{\pi_\theta, \boldsymbol{\pi}_h\}}(\tau_h) \nabla_\theta \log(p_s^{\{\pi_\theta, \boldsymbol{\pi}_h\}}(\tau_h)) \sum_{k=0}^{h-1} \gamma^k r(s_k, a_k)$$

$$= \sum_{\tau_h \in \mathcal{W}_h} p_s^{\{\pi_\theta, \boldsymbol{\pi}_h\}}(\tau_h) \nabla_\theta \log(\pi_\theta(a_0|s_0)) \sum_{k=0}^{h-1} \gamma^k r(s_k, a_k)$$

$$= \mathop{\mathbb{E}}_{\substack{S_0=s, A_0 \sim \pi_\theta(\cdot|S) \\ S_{t+1} \sim p(\cdot|A_t, S_t), A_t \sim \pi_{h-t-1}(\cdot|S_t)}} \Big[ \nabla_\theta \log(\pi^\theta(A_0|S_0)) \sum_{k=0}^{h-1} \gamma^k r(S_k, A_k) \Big]$$

$$= \mathop{\mathbb{E}}_{S_0=s, A_0 \sim \pi_\theta(\cdot|S)} \Big[ \nabla_\theta \log(\pi^\theta(A_0|S_0)) \mathop{\mathbb{E}}_{S_{t+1} \sim p(\cdot|A_t, S_t), A_t \sim \pi_{h-t-1}(\cdot|S_t)} \Big[ \sum_{k=0}^{h-1} \gamma^k r(S_k, A_k) | S_0, A_0 \Big] \Big]$$

$$= \mathop{\mathbb{E}}_{S=s, A \sim \pi_\theta(\cdot|S)} \Big[ \nabla_\theta \log(\pi_\theta(A|S)) Q_{h+1}^{\boldsymbol{\pi}_h}(S, A) \Big].$$

$\square$

### C.5 Performance Difference Lemma

**Lemma C.4.** *Let* $\boldsymbol{\pi}_{h+1}, \boldsymbol{\pi}'_{h+1} \in \Pi^{h+1}$. *Then, it holds that*

$$(i) \quad V_{h+1}^{\boldsymbol{\pi}_{h+1}}(s) - V_{h+1}^{\boldsymbol{\pi}'_{h+1}}(s) = \mathop{\mathbb{E}}_{\substack{S_0=s, A_t \sim \pi_{h-t}(\cdot|S_t) \\ S_{t+1} \sim p(\cdot|S_t, A_t)}} \Big[ \sum_{t=0}^{h} \gamma^t A_{h+1-t}^{\boldsymbol{\pi}'_{h+1-t}}(S_t, A_t) \Big].$$

*(ii) If both policies only differ in the first policy, i.e.* $\pi_h = \pi'_h$, *then the above equation simplifies to*

$$V_{h+1}^{\boldsymbol{\pi}_{h+1}}(s) - V_{h+1}^{\boldsymbol{\pi}'_{h+1}}(s) = \mathop{\mathbb{E}}_{S_0=s, A_0 \sim \pi_h(\cdot|S_0)} \Big[ A_{h+1}^{\boldsymbol{\pi}'_{h+1}}(S_0, A_0) \Big]. \tag{10}$$

*Proof.* A similar proof can be found in [12, Thm A.6]. However, we will need an index shift and additionally to consider the discount factor $\gamma$. First, let $\boldsymbol{\pi}_{h+1}, \boldsymbol{\pi}'_{h+1} \in \Pi^{h+1}$ be two arbitrary policies. We have

$$V_{h+1}^{\boldsymbol{\pi}_{h+1}}(s) - V_{h+1}^{\boldsymbol{\pi}'_{h+1}}(s)$$

$$= \mathop{\mathbb{E}}_{\substack{S_0=s, A_t \sim \pi_{h-t}(\cdot|S_t) \\ S_{t+1} \sim p(\cdot|S_t, A_t)}} \Big[ \sum_{t=0}^{h} \gamma^t r(S_t, A_t) \Big] - V_{h+1}^{\boldsymbol{\pi}'_{h+1}}(s)$$

$$= \mathop{\mathbb{E}}_{\substack{S_0=s, A_t \sim \pi_{h-t}(\cdot|S_t) \\ S_{t+1} \sim p(\cdot|S_t, A_t)}} \Big[ \sum_{t=0}^{h} \gamma^t r(S_t, A_t) + \sum_{t=0}^{h} \gamma^t V_{h+1-t}^{\boldsymbol{\pi}'_{h+1-t}}(S_t) - \sum_{t=0}^{h} \gamma^t V_{h+1-t}^{\boldsymbol{\pi}'_{h+1-t}}(S_t) \Big] - V_{h+1}^{\boldsymbol{\pi}'_{h+1}}(s)$$

$$= \mathop{\mathbb{E}}_{\substack{S_0=s, A_t \sim \pi_{h-t}(\cdot|S_t) \\ S_{t+1} \sim p(\cdot|S_t, A_t)}} \Big[ \sum_{t=0}^{h} \gamma^t r(S_t, A_t) + \sum_{t=1}^{h} \gamma^t V_{h+1-t}^{\boldsymbol{\pi}'_{h+1-t}}(S_t) - \sum_{t=0}^{h} \gamma^t V_{h+1-t}^{\boldsymbol{\pi}'_{h+1-t}}(S_t) \Big]$$

$$= \underset{\substack{S_0=s,A_t\sim\pi_{h-t}(\cdot|S_t) \\ S_{t+1}\sim p(\cdot|S_t,A_t)}}{\mathbb{E}} \left[ \sum_{t=0}^{h} \gamma^t r(S_t, A_t) + \sum_{t=0}^{h-1} \gamma^{t+1} V_{h-t}^{\boldsymbol{\pi}'_{h-t}}(S_{t+1}) - \sum_{t=0}^{h} \gamma^t V_{h+1-t}^{\boldsymbol{\pi}'_{h+1-t}}(S_t) \right]$$

$$= \underset{\substack{S_0=s,A_t\sim\pi_{h-t}(\cdot|S_t) \\ S_{t+1}\sim p(\cdot|S_t,A_t)}}{\mathbb{E}} \left[ \sum_{t=0}^{h} \gamma^t \left( r(S_t, A_t) + \gamma V_{h-t}^{\boldsymbol{\pi}'_{h-t}}(S_{t+1}) - V_{h+1-t}^{\boldsymbol{\pi}'_{h+1-t}}(S_t) \right) \right]$$

$$= \underset{\substack{S_0=s,A_t\sim\pi_{h-t}(\cdot|S_t) \\ S_{t+1}\sim p(\cdot|S_t,A_t)}}{\mathbb{E}} \left[ \sum_{t=0}^{h} \gamma^t \left( Q_{h+1-t}^{\boldsymbol{\pi}'_{h-t}}(S_t, A_t) - V_{h+1-t}^{\boldsymbol{\pi}'_{h+1-t}}(S_t) \right) \right]$$

$$= \underset{\substack{S_0=s,A_t\sim\pi_{h-t}(\cdot|S_t) \\ S_{t+1}\sim p(\cdot|S_t,A_t)}}{\mathbb{E}} \left[ \sum_{t=0}^{h} \gamma^t A_{h+1-t}^{\boldsymbol{\pi}'_{h+1-t}}(S_t, A_t) \right],$$

where in the fifth equation we used the convention $V_0 \equiv 0$, in the sixth equation the definition of the Q-function, and in the last equation the definition of the advantage function. Second, suppose that $\boldsymbol{\pi}_{h+1}$ and $\boldsymbol{\pi}'_{h+1}$ agree on all policies besides $\pi_h$, i.e. $\boldsymbol{\pi}_h = \boldsymbol{\pi}'_h$. Then, for any $t > 0$, it holds that

$$\underset{\substack{S_0=s,A_t\sim\pi_{h-t}(\cdot|S_t) \\ S_{t+1}\sim p(\cdot|S_t,A_t)}}{\mathbb{E}} \left[ A_{h+1-t}^{\boldsymbol{\pi}'_{h+1-t}}(S_t, A_t) \right]$$

$$= \sum_{a\in\mathcal{A}} \pi_h(a|s) \sum_{s'\in\mathcal{S}} p(s'|s,a) \underset{\substack{S_0=s',A_t\sim\pi_{h-t-1}(\cdot|S_t) \\ S_{t+1}\sim p(\cdot|S_t,A_t)}}{\mathbb{E}} \left[ A_{h+1-t}^{\boldsymbol{\pi}'_{h+1-t}}(S_t, A_t) \right]$$

$$= \sum_{a\in\mathcal{A}} \pi_h(a|s) \sum_{s'\in\mathcal{S}} p(s'|s,a) \left( \underset{\substack{S_0=s',A_t\sim\pi'_{h-t-1}(\cdot|S_t) \\ S_{t+1}\sim p(\cdot|S_t,A_t)}}{\mathbb{E}} \left[ Q_{h+1-t}^{\boldsymbol{\pi}'_{h-t}}(S_t, A_t) - V_{h+1-t}^{\boldsymbol{\pi}'_{h+1-t}}(S_t) \right] \right)$$

$$= 0.$$

This proves the claim. $\qquad\square$

### C.6  Convergence under Gradient Domination

**Lemma C.5.** *Let $f : \mathbb{R}^d \to \mathbb{R}$ be L-smooth (i.e. L-Lipschitz continuity of the gradient), $f^* = \sup_x f(x) < \infty$. Denote the gradient ascent procedure by $x_{n+1} = x_n + \alpha\nabla f(x_n)$ for some $x_1 \in \mathbb{R}^d$ and $\alpha = \frac{1}{L}$ and assume that $f$ satisfies the following gradient domination property for some $b > 0$ along the gradient trajectory,*

$$\|\nabla f(x_n)\|_2 \le b(f^* - f(x_n)) \quad \forall n \ge 1.$$

*Then the convergence rate*

$$f^* - f(x_n) \le \frac{2}{\alpha b^2 n},$$

*holds true if $f^* - f(x_1) \le \frac{2}{\alpha b^2}$.*

*Proof.* We apply the descent lemma of smooth functions [4, Lem 5.7] on $-f$ and obtain

$$f(y) \ge f(x) + \nabla f(x)^T (y - x) - \frac{L}{2}\|y - x\|^2.$$

Now for gradient ascent updates we have that

$$f(x_{n+1}) \ge f(x_n) + \nabla f(x_n)^T (x_{n+1} - x_n) - \frac{L}{2}\|x_{n+1} - x_n\|^2$$

$$= f(x_n) + \alpha\|\nabla f(x_n)\|^2 - \frac{L\alpha^2}{2}\|\nabla f(x_n)\|^2$$

$$= f(x_n) + \left(\alpha - \frac{L\alpha^2}{2}\right)\|\nabla f(x_n)\|^2.$$

It follows that

$$f^* - f(x_{n+1}) \le f^* - f(x_n) - \left(\alpha - \frac{L\alpha^2}{2}\right)\|\nabla f(x_n)\|^2.$$

Exploiting the gradient domination along the gradient trajectory results in

$$f^* - f(x_{n+1}) \le f^* - f(x_n) - \left(\alpha - \frac{L\alpha^2}{2}\right)b^2(f^* - f(x_n))^2.$$

As $\alpha = \frac{1}{L}$, we have

$$f^* - f(x_{n+1}) \le f^* - f(x_n) - \frac{b^2}{2L}(f^* - f(n_k))^2.$$

When $f^* - f(x_1) \le \frac{2}{\alpha b^2}$, then $f^* - f(x_n) \le \frac{2}{\alpha b^2 n}$ by [12, Lem. B.7].

$\square$

# D   Proofs of Section 4

## D.1   Proof for Section 4.1

*Proof for Proposition 4.1.* Recall the claim:

$$\|V_\infty^* - V_\infty^{\hat{\pi}_H^*}\|_\infty \le \left\|V_\infty^* - \sup_\theta V_\infty^{\pi_\theta}\right\|_\infty + \left\|\sup_\theta V_\infty^{\pi_\theta} - \sup_{\theta_0,\dots,\theta_{H-1}} V_H^{\{\pi_{\theta_{H-1}},\dots\pi_{\theta_0}\}}\right\|_\infty$$

$$+ \left\|\sup_{\theta_0,\dots,\theta_{H-1}} V_H^{\{\pi_{\theta_{H-1}},\dots\pi_{\theta_0}\}} - V_H^{\hat{\pi}_H^*}\right\|_\infty + \|V_H^{\hat{\pi}_H^*} - V_\infty^{\hat{\pi}_H^*}\|_\infty$$

$$\le \left\|V_\infty^* - \sup_\theta V_\infty^{\pi_\theta}\right\|_\infty + \frac{\gamma^H R^*}{1-\gamma}$$

$$+ \sum_{h=0}^{H-1} \gamma^{H-h-1}\left\|\sup_\theta T^{\pi_\theta}(V_h^{\hat{\pi}_h^*}) - V_{h+1}^{\hat{\pi}_{h+1}^*}\right\|_\infty + \frac{1}{1-\gamma}\|V_{H+1}^{\hat{\pi}_{H+1}^*} - V_H^{\hat{\pi}_H^*}\|_\infty.$$

The first inequality follows directly from the triangle inequality of the supremum norm. Note here that we cannot simplify the first error further as it will depend on the chosen parametrization. Before we prove the second inequality, note that

$$\left|\sup_\theta f(\theta) - \sup_\theta g(\theta)\right| \le \sup_\theta |f(\theta) - g(\theta)|$$

for any two functions $f, g : \mathbb{R}^d \to \mathbb{R}$. If $\sup_\theta f(\theta) - \sup_\theta g(\theta) \ge 0$, then

$$\left|\sup_\theta f(\theta) - \sup_\theta g(\theta)\right| = \sup_\theta f(\theta) - \sup_\theta g(\theta) \le \sup_\theta(f(\theta) - g(\theta)) \le \sup_\theta |f(\theta) - g(\theta)|.$$

If $\sup_\theta f(\theta) - \sup_\theta g(\theta) < 0$, then the role of $f$ and $g$ are swapped. Thus, we have

$$\|\sup_\theta T^{\pi_\theta}(V) - \sup_\theta T^{\pi_\theta}(G)\|_\infty \le \|\sup_\theta (T^{\pi_\theta}(V) - T^{\pi_\theta}(G))\|_\infty \le \gamma\|V - G\|_\infty. \quad (11)$$

We treat the second, third and fourth terms separately.

**For the second term**, we prove that $\left\|\sup_\theta V_\infty^{\pi_\theta} - \sup_{\theta_0,\dots,\theta_{H-1}} V_H^{\{\pi_{\theta_{H-1}},\dots\pi_{\theta_0}\}}\right\|_\infty \le \frac{\gamma^H R^*}{1-\gamma}$ similar
to Lemma C.1. Let $s \in \mathcal{S}$ be arbitrary but fixed. If $\sup_\theta V_\infty^{\pi_\theta}(s) - \sup_{\theta_0,\dots,\theta_{H-1}} V_H^{\{\pi_{\theta_{H-1}},\dots\pi_{\theta_0}\}}(s) \ge 0$, then

$$\sup_\theta V_\infty^{\pi_\theta}(s) - \sup_{\theta_0,\dots,\theta_{H-1}} V_H^{\{\pi_{\theta_{H-1}},\dots\pi_{\theta_0}\}}(s) \le \sup_\theta V_\infty^{\pi_\theta}(s) - \sup_\theta V_H^{\pi_\theta}(s)$$

$$\le \sup_\theta(V_\infty^{\pi_\theta}(s) - V_H^{\pi_\theta}(s))$$

$$= \sup_\theta \mathop{\mathbb{E}}_{\substack{S_0=s,A_t\sim\pi_\theta(\cdot|S_t)\\S_{t+1}\sim p(\cdot|S_t,A_t)}}\left[\sum_{t=H}^\infty \gamma^t r(S_t, A_t)\right]$$

$$\le \sum_{t=H}^\infty \gamma^t R^* = \frac{\gamma^H}{1-\gamma}R^*,$$

where we used (11) in the second inequality. On the other hand, if $\sup_\theta V_\infty^{\pi_\theta}(s) - \sup_{\theta_0,\dots,\theta_{H-1}} V_H^{\{\pi_{\theta_{H-1}},\dots\pi_{\theta_0}\}}(s) < 0$, then

$$\sup_{\theta_0,\dots,\theta_{H-1}} V_H^{\{\pi_{\theta_{H-1}},\dots\pi_{\theta_0}\}}(s) - \sup_\theta V_\infty^{\pi_\theta}(s)$$

$$\leq \sup_{\theta_0,\dots,\theta_{H-1}} V_H^{\{\pi_{\theta_{H-1}},\dots\pi_{\theta_0}\}}(s) - \sup_{\theta_0,\dots,\theta_{H-1}} V_\infty^{(\{\pi_{\theta_{H-1}},\dots\pi_{\theta_0}\})_\infty}(s)$$

$$\leq \sup_{\theta_0,\dots,\theta_{H-1}} \left( V_H^{\{\pi_{\theta_{H-1}},\dots\pi_{\theta_0}\}}(s) - V_\infty^{(\{\pi_{\theta_{H-1}},\dots\pi_{\theta_0}\})_\infty}(s) \right)$$

$$= \sup_{\theta_0,\dots,\theta_{H-1}} - \mathop{\mathbb{E}}_{\substack{S_0=s,A_t\sim\pi_{\theta_{H-(t \bmod H)-1}}(\cdot|S_t) \\ S_{t+1}\sim p(\cdot|S_t,A_t)}} \left[ \sum_{t=H}^{\infty} \gamma^t r(S_t, A_t) \right]$$

$$\leq \sum_{t=H}^{\infty} \gamma^t R^* = \frac{\gamma^H}{1-\gamma} R^*.$$

Collecting these together, we obtain that $\left\| \sup_\theta V_\infty^{\pi_\theta} - \sup_{\theta_0,\dots,\theta_{H-1}} V_H^{\{\pi_{\theta_{H-1}},\dots\pi_{\theta_0}\}} \right\|_\infty \leq \frac{\gamma^H R^*}{1-\gamma}$, since $s \in \mathcal{S}$ was chosen arbitrary.

**For the third term**, we show that

$$\left\| \sup_{\theta_0,\dots,\theta_{H-1}} V_H^{\{\pi_{\theta_{H-1}},\dots\pi_{\theta_0}\}} - V_H^{\hat{\pi}_H^*} \right\|_\infty \leq \sum_{h=0}^{H-1} \gamma^{H-h-1} \|\sup_\theta T^{\pi_\theta}(V_h^{\hat{\pi}_h^*}) - T^{\hat{\pi}_h^*}(V_h^{\hat{\pi}_h^*})\|_\infty \qquad (12)$$

holds for all $H \geq 1$ by induction. For $H = 1$ we have by (5) that

$$\|\sup_\theta V_1^{\pi_\theta} - V_1^{\hat{\pi}_1^*}\|_\infty = \left\| \sup_\theta T^{\pi_\theta}(V_0^{\hat{\pi}_0^*}) - T^{\hat{\pi}_1^*}(V_0^{\hat{\pi}_0^*}) \right\|_\infty,$$

with the convention $V_0 \equiv 0$ and $\hat{\pi}_0^* = \emptyset$. So assume that Equation (12) holds for some $H \geq 1$; then for $H+1$ we have

$$\left\| \sup_{\theta_0,\dots,\theta_H} V_{H+1}^{\{\pi_{\theta_H},\dots\pi_{\theta_0}\}} - V_{H+1}^{\hat{\pi}_{H+1}^*} \right\|_\infty$$

$$= \left\| \sup_{\theta_H} T^{\pi_{\theta_H}} \left( \sup_{\theta_0,\dots,\theta_{H-1}} V_H^{\{\pi_{\theta_{H-1}},\dots\pi_{\theta_0}\}} \right) - T^{\hat{\pi}_H^*}(V_H^{\hat{\pi}_H^*}) \right\|_\infty$$

$$\leq \left\| \sup_{\theta_H} T^{\pi_{\theta_H}} \left( \sup_{\theta_0,\dots,\theta_{H-1}} V_H^{\{\pi_{\theta_{H-1}},\dots\pi_{\theta_0}\}} \right) - \sup_{\theta_H} T^{\pi_{\theta_H}}(V_H^{\hat{\pi}_H^*}) \right\|_\infty$$

$$+ \left\| \sup_{\theta_H} T^{\pi_{\theta_H}}(V_h^{\hat{\pi}_H^*}) - T^{\hat{\pi}_H^*}(V_H^{\hat{\pi}_H^*}) \right\|_\infty$$

$$\leq \gamma \left\| \sup_{\theta_0,\dots,\theta_{H-1}} V_H^{\{\pi_{\theta_{H-1}},\dots\pi_{\theta_0}\}} - V_H^{\hat{\pi}_H^*} \right\|_\infty + \left\| \sup_{\theta_H} T^{\pi_{\theta_H}}(V_H^{\hat{\pi}_H^*}) - T^{\hat{\pi}_H^*}(V_H^{\hat{\pi}_H^*}) \right\|_\infty$$

$$\leq \gamma \sum_{h=0}^{H-1} \gamma^{H-h-1} \|\sup_\theta T^{\pi_\theta}(V_h^{\hat{\pi}_h^*}) - T^{\hat{\pi}_h^*}(V_h^{\hat{\pi}_h^*})\|_\infty + \left\| \sup_\theta T^{\pi_\theta}(V_H^{\hat{\pi}_H^*}) - T^{\hat{\pi}_H^*}(V_H^{\hat{\pi}_H^*}) \right\|_\infty$$

$$= \sum_{h=0}^{H} \gamma^{H-h} \|\sup_\theta T^{\pi_\theta}(V_h^{\hat{\pi}_h^*}) - T^{\hat{\pi}_h^*}(V_h^{\hat{\pi}_h^*})\|_\infty,$$

where we used (5), as well as (11), and the induction assumption. This yields the desired claim (12) for all $H \geq 1$.

**For the fourth error term**, we have to deal with the error of applying the final policy as stationary policy. We use Lemma C.2 to arrive at

$$\|V_H^{\hat{\pi}_H^*} - V_\infty^{\hat{\pi}_H^*}\|_\infty \leq \frac{1}{1-\gamma} \|T^{\hat{\pi}_H^*}(V_H^{\hat{\pi}_H^*}) - V_H^{\hat{\pi}_H^*}\|_\infty$$

$$= \frac{1}{1-\gamma} \|V_{H+1}^{\hat{\pi}_{H+1}^*} - V_H^{\hat{\pi}_H^*}\|_\infty,$$

where we used again (5) in the last line. $\qquad\square$

*Proof of Corollary 4.3.* Adjusting (7) to zero approximation error and exploiting Assumption 4.2 we obtain

$$\|V_\infty^* - V_H^{\hat{\boldsymbol{\pi}}_H^*}\|_\infty \leq \frac{\gamma^H R^*}{1-\gamma} + \sum_{h=0}^{H-1} \gamma^{H-h-1}\epsilon_h \to 0 \quad \text{for } H \to \infty.$$

For the second part of the theorem note that $V_H^{\boldsymbol{\pi}_H^*}$ converges in the supremum norm to $V_\infty^*$ by the first part. This implies that $V_H^{\boldsymbol{\pi}_H^*}$ is a Cauchy sequence with respect to the supremum norm, i.e. $\|V_{H+1}^{\boldsymbol{\pi}_{H+1}^*} - V_H^{\boldsymbol{\pi}_H^*}\|_\infty \to 0$ for $H \to \infty$. The claim follows directly from Proposition 4.1. □

## D.2 Proof of Theorem 4.6

Before we prove Theorem 4.6, we will discuss the similarities and differences between DynPG and the finite-time Dynamic Policy Gradient (FT-DynPG) Algorithm proposed in [12].

1. FT-DynPG has a prefixed time horizon $H$ and trains policy backwards in time. In contrast, DynPG adds arbitrarily many policies in the beginning and can therefore be applied without prefixed $H$.

2. Given a fixed time horizon $H$, FT-DynPG returns a non-stationary policy $\hat{\boldsymbol{\pi}}_H^*$ for an $H$-step MDP, but without discounting $\gamma = 1$. When DynPG is run for the same fixed number of iterations $H$, then FT-DynPG and DynPG only differ by the discount factor: The value functions $V_{h,H}$ defined in [12] for FT-DynPG are by the strong Markov property equivalent to $V_{0,H-h}$ via index shifting. The function $V_{0,H-h}$ differs from our $H-h$-step value function $V_{H-h}$ defined in (1) only by the discount factor $\gamma$.

In order to prove Theorem 4.6, we have to adapt the proof for [12, Lem. 3.4] to an additional discount factor. Note, that we cannot use [16, Thm. 2] for bandits, as we consider contextual bandits. Further, we cannot use [16, Thm. 4] with $\gamma = 0$ as they just consider positive rewards in $[0, 1]$ which is inconvenient for our contextual bandit setting where the maximal rewards grows when we add a new time-epoch in the beginning.

Therefore, we will adapt the proofs in [16, 12] and start with deriving the smoothness of our objective functions.

**Lemma D.1.** *Suppose $\pi_\theta = \text{softmax}(\theta)$. Then, for arbitrary $\boldsymbol{\pi}_h \in \Pi^h$ and $\mu \in \Delta(\mathcal{S})$, the function $\theta \mapsto V_{h+1}^{\{\pi_\theta, \boldsymbol{\pi}_h\}}(\mu)$ is $L_h$-smooth with $L_h = \frac{2R^*(1-\gamma^{h+1})}{(1-\gamma)}$.*

*Proof.* The proof is similar to [12, Lem. B.8]. Note that we can interpret $V_{h+1}^{\{\pi_\theta, \boldsymbol{\pi}_h\}}(\mu)$ as a contextual bandit problem, i.e. a discounted (infinite-time) MDP with discount factor 0. The reward of the contextual bandit problem is almost surely bounded in $[-\frac{1-\gamma^{h+1}}{1-\gamma}R^*, \frac{1-\gamma^{h+1}}{1-\gamma}R^*]$, because

$$\sum_{t=0}^h \gamma^t R^* = \frac{1-\gamma^{h+1}}{1-\gamma} R^*.$$

We can apply [31, Lem. 4.4 and Lem. 4.8] with $R_{\max} = \frac{1-\gamma^{h+1}}{1-\gamma}R^*$, $G^2 = 1 - \frac{1}{|\mathcal{A}|} \leq 1$ and $F = 1$ to obtain the smoothness constant $L_h = \frac{2R^*(1-\gamma^{h+1})}{(1-\gamma)}$. □

Second, we obtain the following gradient domination property.

**Lemma D.2.** *Under Assumption 4.5 it holds for any $\boldsymbol{\pi}_h \in \Pi^h$ that*

$$\|\nabla_\theta V_{h+1}^{\{\pi_\theta, \boldsymbol{\pi}_h\}}(\mu)\| \leq \min_{s \in \mathcal{S}} \pi_\theta(a^*(s)|s)\big(T^*(V_h^{\boldsymbol{\pi}_h})(\mu) - V_{h+1}^{\{\pi_\theta, \boldsymbol{\pi}_h\}}(\mu)\big),$$

*where $a^*(s)$ denotes the (unique) action taken after the greedy policy $\pi^{V_h^{\boldsymbol{\pi}_h}}$.*

**Remark D.3.** *Without loss of generality, we assume that the action $a^*(s)$ is unique for any fixed future policy $\boldsymbol{\pi}_h$, otherwise one can consider $\min_{a \text{ optimal action in } s} \pi_\theta(a|s)$.*

*Proof.* First note from Theorem C.3 that under the tabular softmax parametrization we have

$$\nabla_\theta V_{h+1}^{\{\pi_\theta, \boldsymbol{\pi}_h\}}(\mu) = \sum_{s\in\mathcal{S}} \mu(s) \underset{S=s, A\sim\pi_\theta(\cdot|S)}{\mathbb{E}} \left[\nabla_\theta \log(\pi_\theta(A|S)) Q_{h+1}^{\boldsymbol{\pi}_h}(S,A)\right].$$

Hence, with the derivative of the softmax function

$$\frac{\partial \log(\pi_\theta(a|s))}{\partial\theta(s',a')} = \mathbf{1}_{\{s=s'\}}\left(\mathbf{1}_{\{a=a'\}} - \pi_\theta(a'|s)\right)$$

it holds that

$$
\begin{aligned}
\frac{\partial V_{h+1}^{\{\pi_\theta, \boldsymbol{\pi}_h\}}(s)}{\partial\theta(s',a')} &= \mathbf{1}_{\{s=s'\}} \underset{S=s', A\sim\pi_\theta(\cdot|S)}{\mathbb{E}} \left[\left(\mathbf{1}_{\{A=a'\}} - \pi_\theta(a'|s')\right) Q_{h+1}^{\boldsymbol{\pi}_h}(S,A)\right]\\
&= \mathbf{1}_{\{s=s'\}}\left(\pi_\theta(a'|s')Q_{h+1}^{\boldsymbol{\pi}_h}(s',a') - \pi_\theta(a'|s') \underset{S=s', A\sim\pi_\theta(\cdot|S)}{\mathbb{E}} \left[Q_{h+1}^{\boldsymbol{\pi}_h}(S,A)\right]\right)\\
&= \mathbf{1}_{\{s=s'\}}\pi_\theta(a'|s')\left(Q_{h+1}^{\boldsymbol{\pi}_h}(s',a') - V_{h+1}^{\{\pi_\theta,\boldsymbol{\pi}_h\}}(s')\right)\\
&= \mathbf{1}_{\{s=s'\}}\pi_\theta(a'|s')A_{h+1}^{\{\pi_\theta,\boldsymbol{\pi}_h\}}(s',a').
\end{aligned}
\tag{13}
$$

We deduce from Lemma C.4, Equation (10), that

$$T^*(V_h^{\boldsymbol{\pi}_h})(s) - T^{\pi_h}(V_h^{\boldsymbol{\pi}_h})(s) = \underset{S_0=s, A_0\sim\pi_h^*(\cdot|S_0)}{\mathbb{E}}\left[A_{h+1}^{\boldsymbol{\pi}_{h+1}}(S_0,A_0)\right].$$

Finally, the rest of the proof is derived as in [12, Lem. B.9]

$$
\begin{aligned}
\|\nabla_\theta V_{h+1}^{\{\pi_\theta,\boldsymbol{\pi}_h\}}(\mu)\|_2 &= \left\|\sum_{s\in\mathcal{S}}\mu(s)\frac{\partial V_{h+1}^{\{\pi_\theta,\boldsymbol{\pi}_h\}}(s)}{\partial\theta}\right\|_2\\
&= \left[\sum_{s'\in\mathcal{S}}\sum_{a'\in\mathcal{A}}\left(\sum_{s\in\mathcal{S}}\mu(s)\frac{\partial V_{h+1}^{\{\pi_\theta,\boldsymbol{\pi}_h\}}(s)}{\partial\theta(s',a')}\right)^2\right]^2\\
&= \left[\sum_{a'\in\mathcal{A}}\left(\sum_{s\in\mathcal{S}}\mu(s)\pi_\theta(a'|s)A_{h+1}^{\{\pi_\theta,\boldsymbol{\pi}_h\}}(s,a')\right)^2\right]^2\\
&\geq \left|\sum_{s\in\mathcal{S}}\mu(s)\pi_\theta(a^*(s)|s)A_{h+1}^{\{\pi_\theta,\boldsymbol{\pi}_h\}}(s,a^*(s))\right|\\
&= \left|\sum_{s\in\mathcal{S}}\mu(s)\pi_\theta(a^*(s)|s)\underset{S=s, A\sim\pi^{V_h^{\boldsymbol{\pi}_h}}(\cdot|s)}{\mathbb{E}}\left[A_{h+1}^{\{\pi_\theta,\boldsymbol{\pi}_h\}}(S,A)\right]\right|\\
&= \sum_{s\in\mathcal{S}}\mu(s)\pi_\theta(a^*(s)|s)\left(T^*(V_h^{\boldsymbol{\pi}_h})(s) - T^{\pi_\theta}(V_h^{\boldsymbol{\pi}_h})(s)\right)\\
&\geq \min_{s\in\mathcal{S}}\pi_\theta(a^*(s)|s)\left(T^*(V_h^{\boldsymbol{\pi}_h})(\mu) - T^{\pi_\theta}(V_h^{\boldsymbol{\pi}_h})(\mu)\right),
\end{aligned}
$$

where $a^*(s)$ denotes the action taken after the greedy policy $\pi^{V_h^{\boldsymbol{\pi}_h}}$ (c.f. Remark D.3). $\qquad\square$

The next step is to show that the term $\min_{s\in\mathcal{S}}\pi_\theta(a^*(s)|s)$ can be bounded (uniformly in $s\in\mathcal{S}$) from below by $\frac{1}{|\mathcal{A}|}$ along the gradient ascent trajectory, when softmax is initialized uniformly.

**Lemma D.4.** *Let Assumption 4.2 hold and denote by $(\theta_n)_{n\geq 0}$ the gradient ascent sequence in epoch $h\geq 0$ of DynPG under the fixed future policy $\hat{\boldsymbol{\pi}}_h^*\in\Pi^h$. Suppose further that $\eta_h = \frac{1-\gamma}{2R^*(1-\gamma^{h+1})}$ is chosen as in Theorem 4.6. Then, for any $s\in\mathcal{S}$, it holds that*

$$\min_{n\geq 0}\pi_{\theta_n}(a^*(s)|s) = \frac{1}{|\mathcal{A}|}.$$

*where $a^*(s)$ denotes the (unique) action taken after the greedy policy $\pi^{V_h^{\hat{\boldsymbol{\pi}}_h^*}}$ (see Remark D.3).*

*Proof.* The proof is adapted from [16, Lem. 5] and [12, Lem. B.10, Prop. 3.3].
First, we define the sets

$$\mathcal{R}_1(s) = \{\theta : \pi_\theta(a^*(s)|s) \geq \pi_\theta(a|s) \forall a \neq a^*(s)\}$$

$$\mathcal{R}_2(s) = \left\{\theta : \frac{\partial V_{h+1}^{\{\pi_\theta, \hat{\boldsymbol{\pi}}_h^*\}}(s)}{\partial \theta(s, a^*(s))} \geq \frac{\partial V_{h+1}^{\{\pi_\theta, \hat{\boldsymbol{\pi}}_h^*\}}(s)}{\partial \theta(s, a^*(s))} \forall a \neq a^*(s)\right\}.$$

**Claim 1:** It holds that $\theta_n \in \mathcal{R}_2 \implies \theta_{n+1} \in \mathcal{R}_2$.
To prove this claim, we first deduce from (13) that

$$\frac{\partial V_{h+1}^{\{\pi_\theta, \hat{\boldsymbol{\pi}}_h^*\}}(s)}{\partial \theta(s, a^*(s))} \geq \frac{\partial V_{h+1}^{\{\pi_\theta, \hat{\boldsymbol{\pi}}_h^*\}}(s)}{\partial \theta(s, a^*(s))} \tag{14}$$

$$\iff \pi_{\theta_n}(a^*(s)|s) A_{h+1}^{\{\pi_\theta, \hat{\boldsymbol{\pi}}_h^*\}}(s, a^*(s)) \geq \pi_{\theta_n}(a|s) A_{h+1}^{\{\pi_\theta, \hat{\boldsymbol{\pi}}_h^*\}}(s, a).$$

Let $a \neq a^*(s)$ be arbitrary. We consider two cases to proof claim 1:

1. Suppose that $\pi_{\theta_n}(a^*(s)|s) \geq \pi_{\theta_n}(a|s)$ for any $a \neq a^*(s)$. Then, is holds that $\theta_n(s, a^*(s)) \geq \theta_n(s, a)$ by the definition of softmax. Next, as $\theta_n \in \mathcal{R}_2^*$, we derive

$$\theta_{n+1}(s, a^*(s)) = \theta_n(s, a^*(s)) + \eta_h \mu_h(s) \frac{\partial V_{h+1}^{\{\pi_{\theta_n}, \hat{\boldsymbol{\pi}}_h^*\}}(s)}{\partial \theta_n(s, a^*(s))}$$

$$\geq \theta_n(s, a) + \eta_h \mu_h(s) \frac{\partial V_{h+1}^{\{\pi_\theta, \hat{\boldsymbol{\pi}}_h^*\}}(s)}{\partial \theta(s, a^*(s))}$$

$$= \theta_n(s, a).$$

Thus, by the definition of the softmax function, $\pi_{\theta_{n+1}}(a^*(s)|s) \geq \pi_{\theta_{n+1}}(a|s)$ for any $a \neq a^*(s)$. Further, because $a^*(s)$ is the greedy action, we arrive at

$$\pi_{\theta_{n+1}}(a^*(s)|s) A_{h+1}^{\{\pi_{\theta_{n+1}}, \hat{\boldsymbol{\pi}}_h^*\}}(s, a^*(s)) \geq \pi_{\theta_n}(a|s) A_{h+1}^{\{\pi_{\theta_{n+1}}, \hat{\boldsymbol{\pi}}_h^*\}}(s, a),$$

such that $\theta_{n+1} \in \mathcal{R}_2(s)$ by (14).

2. Suppose that $\pi_{\theta_n}(a^*(s)|s) < \pi_{\theta_n}(a|s)$ for any $a \neq a^*(s)$. Then, $\theta_n(s, a^*(s)) - \theta_n(s, a) < 0$ by definition of the softmax function. As $\theta_n \in \mathcal{R}_2(s)$, it holds that

$$\theta_{n+1}(s, a^*(s)) - \theta_{n+1}(s, a)$$

$$\geq \theta_n(s, a^*(s)) + \eta_h \mu(s) \frac{\partial V_{h+1}^{\{\pi_\theta, \hat{\boldsymbol{\pi}}_h^*\}}(s)}{\partial \theta(s, a^*(s))} - \theta_n(s, a) - \eta_h \mu(s) \frac{\partial V_{h+1}^{\{\pi_\theta, \hat{\boldsymbol{\pi}}_h^*\}}(s)}{\partial \theta(s, a)}$$

$$\geq \theta_n(s, a^*(s)) - \theta_n(s, a).$$

Thus, we obtain

$$(1 - \exp(\theta_{n+1}(s, a^*(s)) - \theta_{n+1}(s, a))) \leq (1 - \exp(\theta_n(s, a^*(s)) - \theta_n(s, a))) < 1.$$

By the ascent lemma for smooth functions [16, Lem. 18] it follows monotonicity in the objective function (due to small enough step size $\eta_h = \frac{1}{L_h}$ with $L_h$ the smoothness constant in Lemma D.1) such that $V_{h+1}^{\{\pi_{\theta_{n+1}}, \hat{\boldsymbol{\pi}}_h^*\}}(s) \geq V_{h+1}^{\{\pi_{\theta_n}, \hat{\boldsymbol{\pi}}_h^*\}}(s)$. So,

$$(1 - \exp(\theta_{n+1}(s, a^*(s)) - \theta_{n+1}(s, a)))\left(Q_{h+1}^{\hat{\boldsymbol{\pi}}_h^*}(s, a^*(s)) - V_{h+1}^{\{\pi_{\theta_{n+1}}, \hat{\boldsymbol{\pi}}_h^*\}}(s)\right)$$

$$\leq (1 - \exp(\theta_n(s, a^*(s)) - \theta_n(s, a)))\left(Q_{h+1}^{\hat{\boldsymbol{\pi}}_h^*}(s, a^*(s)) - V_{h+1}^{\{\pi_{\theta_n}, \hat{\boldsymbol{\pi}}_h^*\}}(s)\right)$$

We rearrange (14) and obtain that $\theta \in \mathcal{R}_2(s)$ is equivalent to

$$Q_{h+1}^{\hat{\boldsymbol{\pi}}_h^*}(s, a^*(s)) - Q_{h+1}^{\hat{\boldsymbol{\pi}}_h^*}(s, a) \geq (1 - \exp(\theta_n(s, a^*(s)) - \theta_n(s, a))) A_{h+1}^{\{\pi_\theta, \hat{\boldsymbol{\pi}}_h^*\}}(s, a^*(s)).$$

We deduce by $\theta_n \in \mathcal{R}_2(s)$ that

$$(1 - \exp(\theta_{n+1}(s, a^*(s)) - \theta_{n+1}(s, a)))\Big(Q_{h+1}^{\hat{\boldsymbol{\pi}}_h^*}(s, a^*(s)) - V_{h+1}^{\{\pi_{\theta_{n+1}}, \hat{\boldsymbol{\pi}}_h^*\}}(s)\Big)$$

$$\leq (1 - \exp(\theta_n(s, a^*(s)) - \theta_n(s, a)))\Big(Q_{h+1}^{\hat{\boldsymbol{\pi}}_h^*}(s, a^*(s)) - V_{h+1}^{\{\pi_{\theta_n}, \hat{\boldsymbol{\pi}}_h^*\}}(s)\Big)$$

$$\leq Q_{h+1}^{\hat{\boldsymbol{\pi}}_h^*}(s, a^*(s)) - Q_{h+1}^{\hat{\boldsymbol{\pi}}_h^*}(s, a),$$

and thus $\theta_{n+1} \in \mathcal{R}_2(s)$.

This proves claim 1.
**Claim 2:** If $\theta_n \in \mathcal{R}_2(s)$, then it holds that $\pi_{\theta_{n+1}}(a^*(s)|s) \geq \pi_{\theta_n}(a^*(s)|s)$.
Line by line as in Claim 2 of [12, Lem. B.10]. **Claim 3:** It holds that $\theta_n \in \mathcal{R}_1(s) \implies \theta_n \in \mathcal{R}_2(s)$.
Let $\theta_n \in \mathcal{R}_1(s)$. As $a^*(s)$ is optimal we have for any $a \neq a^*(s)$ that

$$A_{h+1}^{\{\pi_{\theta_n}, \hat{\boldsymbol{\pi}}_h^*\}}(s, a^*(s)) \geq A_{h+1}^{\{\pi_{\theta_n}, \hat{\boldsymbol{\pi}}_h^*\}}(s, a).$$

Further by $\theta_n \in \mathcal{R}_1(s)$ it holds

$$\pi_{\theta_n}(a^*(s)|s) A_{h+1}^{\{\pi_{\theta_n}, \hat{\boldsymbol{\pi}}_h^*\}}(s, a^*(s)) \geq \pi_{\theta_n}(a|s) A_{h+1}^{\{\pi_{\theta_n}, \hat{\boldsymbol{\pi}}_h^*\}}(s, a).$$

Hence, by (13), we deduce that $\theta_n \in \mathcal{R}_2(s)$.
**Conclusion of the proof by combining claim 1, claim 2 and claim 3:**
Since $\theta_0$ is initialized such that softmax is the uniform distribution, we have that $\theta_0 \in \mathcal{R}_1(s)$ for all $s \in \mathcal{S}$. By claim 3, we have that $\theta_0 \in \mathcal{R}_2(s)$ and by claim 1 it follows that $\theta_n \in \mathcal{R}_2(s)$ for all $n \geq 0$ and all $s \in \mathcal{S}$. Finally, by claim 2, it follows that $\min_{n \geq 0} \pi_{\theta_n}(s, a^*(s)) = \pi_{\theta_0}(s, a^*(s)) = \frac{1}{|\mathcal{A}|}$ for any $s \in \mathcal{S}$. $\qquad\square$

We finally we collect all preliminary results to prove Theorem 4.6.

*Proof of Theorem 4.6.* First, note that the tabular softmax parametrization can approximate any deterministic policy arbitrarily well. As optimal policies in finite horizon MDPs are deterministic, we have that $\sup_{\theta \in \mathbb{R}^{|\mathcal{S}||\mathcal{A}|}} V_{h+1}^{\{\pi_\theta, \hat{\boldsymbol{\pi}}_h^*\}}(s) = T^*(V_h^{\hat{\boldsymbol{\pi}}_h^*})(s)$ for all $s \in \mathcal{S}$ (zero approximation error induced by the parametrization). Moreover, for any $s \in \mathcal{S}$

$$T^*(V_h^{\hat{\boldsymbol{\pi}}_h^*})(s) - T^{\hat{\pi}_h^*}(V_h^{\hat{\boldsymbol{\pi}}_h^*})(s) = \sum_{s' \in \mathcal{S}} \mu(s') \frac{\mathbf{1}_{s'=s}}{\mu(s')}\Big(T^*(V_h^{\hat{\boldsymbol{\pi}}_h^*})(s') - T^{\hat{\pi}_h^*}(V_h^{\hat{\boldsymbol{\pi}}_h^*})(s')\Big)$$

$$\leq \Big\|\frac{1}{\mu}\Big\|_\infty \Big(T^*(V_h^{\hat{\boldsymbol{\pi}}_h^*})(\mu) - T^{\hat{\pi}_h^*}(V_h^{\hat{\boldsymbol{\pi}}_h^*})(\mu)\Big).$$

For any $\theta \in \mathbb{R}^d$ it holds that

$$T^*(V_h^{\hat{\boldsymbol{\pi}}_h^*})(s) - T^{\pi_\theta}(V_h^{\hat{\boldsymbol{\pi}}_h^*})(s) \leq 2\frac{1 - \gamma^{h+1}}{1 - \gamma} R^* = L_h = \frac{1}{\eta_h} \leq \frac{2|\mathcal{A}|^2}{\eta_h}.$$

Moreover, by Lemma D.1 the function $\theta \mapsto V_{h+1}^{\{\pi_\theta, \hat{\boldsymbol{\pi}}_h^*\}}(\mu)$ is $L_h$-smooth and fulfills the gradient domination property along the gradient ascent steps with $b = \frac{1}{|\mathcal{A}|}$ (Lemma D.2 and Lemma D.4). Hence, we can apply Lemma C.5 with $b = \frac{1}{|\mathcal{A}|}$, $\alpha = \eta_h = \frac{1-\gamma}{2R^*(1-\gamma^{h+1})}$ and $n = N_h$, such that

$$T^*(V_h^{\hat{\boldsymbol{\pi}}_h^*})(\mu) - T^{\pi_{\theta_{N_h}}}(V_h^{\hat{\boldsymbol{\pi}}_h^*})(\mu) \leq \frac{4|\mathcal{A}|^2 R^*(1 - \gamma^{h+1})}{(1 - \gamma)N_h}.$$

By the choice of $N_h$ we deduce for any $s \in \mathcal{S}$

$$T^*(V_h^{\hat{\boldsymbol{\pi}}_h^*})(s) - T^{\hat{\pi}_h^*}(V_h^{\hat{\boldsymbol{\pi}}_h^*})(s) \leq \Big\|\frac{1}{\mu}\Big\|_\infty \Big(T^*(V_h^{\hat{\boldsymbol{\pi}}_h^*})(\mu) - T^{\pi_{\theta_{N_h}}}(V_h^{\hat{\boldsymbol{\pi}}_h^*})(\mu)\Big) \leq \epsilon_h,$$

which proves the claim. $\qquad\square$

## D.3 Proof of Theorem 4.8

We will treat the two cases of value function error and overall error separately and start with the value function error.

**Value function error.** Note that under the tabular softmax parametrization we have zero approximation error and the can upper bound in (7) simplifies to

$$\|V_\infty^* - V_H^{\hat{\boldsymbol{\pi}}_H^*}\|_\infty \leq \frac{\gamma^H R^*}{1-\gamma} + \sum_{h=0}^{H-1} \gamma^{H-h-1}\epsilon_h,$$

where $\epsilon_h$ are upper bounds on one step optimization errors $\|\sup_{\theta \in \mathbb{R}^d} T^{\pi_\theta}(V_h^{\hat{\boldsymbol{\pi}}_h^*}) - V_{h+1}^{\hat{\boldsymbol{\pi}}_{h+1}^*}\|_\infty$ according to Theorem 4.6.

In order to minimize the accumulated number of gradient steps, we have to solve the optimization problem:

$$\min_{H,(\epsilon_h)_{h=0}^{H-1},\epsilon_h>0} \sum_{h=0}^{H-1} N_h = \frac{4R^*|\mathcal{A}|^2}{(1-\gamma)}\left\|\frac{1}{\mu}\right\|_\infty \sum_{h=0}^{H-1} \frac{(1-\gamma^{h+1})}{\epsilon_h}$$

$$\text{subject to} \quad \frac{\gamma^H R^*}{1-\gamma} + \sum_{h=0}^{H-1} \gamma^{H-h-1}\epsilon_h \leq \epsilon. \tag{15}$$

In [28, Sec. 3.2] a similar optimization problem was considered and it was shown that asymptotically (as $\epsilon \to 0$) it suffices to bound both error terms in the constraint by $\frac{\epsilon}{2}$. The first condition, i.e. $\frac{\gamma^H R^*}{1-\gamma} \leq \frac{\epsilon}{2}$, leads to the criterion $H \geq \log_\gamma\left(\frac{(1-\gamma)\epsilon}{2R^*}\right)$. To minimize the number of gradient steps we fix $H = \left\lceil \log_\gamma\left(\frac{(1-\gamma)\epsilon}{2R^*}\right)\right\rceil$. It remains to solve the following optimization problem

$$\min_{(\epsilon_h)_{h=0}^{H-1},\epsilon_h>0} \sum_{h=0}^{H-1} a_h\epsilon_h^{-1} \quad \text{subject to} \quad \sum_{h=0}^{H-1} \gamma^{H-h-1}\epsilon_h \leq \frac{\epsilon}{2}, \tag{16}$$

with $a_h = \left\|\frac{4R^*|\mathcal{A}|^2}{1-\gamma}\frac{1}{\mu}\right\|_\infty (h+1)(1-\gamma^{h+1})$.

We provide a solution of the optimization problem (16) by solving the following more general one:

**Lemma D.5.** *Fix $H > 0$. Let $(a_h)_{h=0}^{H-1}$ and $(b_h)_{h=0}^{H-1}$ be strictly positive sequences. For any $d > 0$ the optimization problem*

$$\min_{(c_h)_{h=0}^{H-1}} \sum_{h=0}^{H-1} a_h c_h^{-1} \quad \text{subject to} \quad \sum_{h=0}^{H-1} b_h c_h \leq d,$$

*is optimally solved for $c_h = C_{d,H}\left(\frac{b_h}{a_h}\right)^{-\frac{1}{2}}$, with $C_{H,d} = d\left(\sum_{h=0}^{H-1}(a_h b_h)^{\frac{1}{2}}\right)^{-1}$.*

*Hence, the minimum of the optimization problem is given by $\frac{1}{d}\left(\sum_{h=0}^{H-1}(a_h b_h)^{\frac{1}{2}}\right)^2$.*

*Proof.* The result and the proof is inspired by [28, Lem. 3.8].
We solve the constrained optimization problem by employing the Lagrange method, i.e

$$\min_{\lambda,(c_h)_{h=0}^{H-1}} \sum_{h=0}^{H-1} a_h c_h^{-1} + \lambda\left(\sum_{h=0}^{H-1} b_h c_h - d\right).$$

The first order conditions are given by

$$-a_h c_h^{-2} + \lambda b_h = 0 \quad \forall h = 0, \ldots, H-1, \quad \text{and} \quad \sum_{h=0}^{H-1} b_h c_h - d = 0.$$

We deduce from the first equations, that there exists a constant $C_{H,d}$ such that $c_h = C_{H,d}\left(\frac{b_h}{a_h}\right)^{-\frac{1}{2}}$ for all $h = 0, \ldots, H-1$. Using this in the second equation we can solve for the constant

$$C_{H,d} = d\Big(\sum_{h=0}^{H-1} b_h^{\frac{1}{2}} a_h^{\frac{1}{2}}\Big)^{-1}.$$

Using the minima $c_h = C_{H,d}\left(\frac{b_h}{a_h}\right)^{-\frac{1}{2}}$ in the optimization function $\sum_{h=0}^{H-1} a_h c_h^{-1}$, we obtain the minimum

$$\sum_{h=0}^{H-1} a_h c_h^{-1} = \sum_{h=0}^{H-1} a_h C_{H,d}^{-1}\left(\frac{b_h}{a_h}\right)^{\frac{1}{2}} = C_{H,\epsilon}^{-1}\sum_{h=0}^{H-1}(a_h b_h)^{\frac{1}{2}} = \frac{1}{d}\left(\sum_{h=0}^{H-1}(a_h b_h)^{\frac{1}{2}}\right)^2.$$

$\square$

Finally, we are ready to state the detailed version of Theorem 4.8 for the value function error.

**Theorem D.6.** *[Detailed version of Theorem 4.8 (2.)]*
*Let Assumption 4.5 hold true and the gradient be accessed exactly. Choose $\epsilon > 0$ and set*

$$H = \left\lceil \log_\gamma\left(\frac{(1-\gamma)\epsilon}{2R^*}\right)\right\rceil,$$

$$\epsilon_h = \frac{\epsilon}{2}\Big(\sum_{t=0}^{H-1}((1-\gamma^{t+1})\gamma^{H-t-1})^{\frac{1}{2}}\Big)^{-1}\left(\frac{\gamma^{H-h-1}}{(1-\gamma^{h+1})}\right)^{-\frac{1}{2}},$$

$$\eta_h = \frac{1-\gamma}{2R^*(1-\gamma^{h+1})}$$

$$N_h = \left\lceil \frac{4R^*(1-\gamma^{h+1})|\mathcal{A}|^2}{(1-\gamma)\epsilon_h}\Big\|\frac{1}{\mu}\Big\|_\infty\right\rceil,$$

*for $h = 0, \ldots H-1$. Then, the non-stationary policy $\hat{\pi}_H^*$ obtained by DynPG under exact gradients achieves $\|V_\infty^* - V_H^{\hat{\pi}_H^*}\|_\infty \leq \epsilon$. The total number of gradient steps are given by*

$$\sum_{h=0}^{H-1} N_h = \frac{8R^*|\mathcal{A}|^2}{(1-\gamma)^2\epsilon}\left\lceil\frac{\log(2R^*(1-\gamma)^{-1}\epsilon^{-1})}{\log(\gamma^{-1})}\right\rceil\Big\|\frac{1}{\mu}\Big\|_\infty.$$

*Proof.* First, note that the choice of $\epsilon_h$ in the theorem is the solution of the optimization problem

$$\min_{(\epsilon_h)} \frac{4R^*|\mathcal{A}|^2}{(1-\gamma)}\Big\|\frac{1}{\mu}\Big\|_\infty \sum_{h=0}^{H-1}\frac{(1-\gamma^{h+1})}{\epsilon_h} \quad \text{subject to} \quad \sum_{h=0}^{H-1}\gamma^{H-h-1}\epsilon_h \leq \frac{\epsilon}{2}.$$

Therefore, choose $a_h = (1-\gamma^{h+1})$, $b_h = \gamma^{H-h-1}$, $c_h = \epsilon_h$ and $d = \frac{\epsilon}{2}$ in Lemma D.5, where we excluded the constant $\frac{4R^*|\mathcal{A}|^2}{(1-\gamma)}\Big\|\frac{1}{\mu}\Big\|_\infty$. Then,

$$\epsilon_h = C_{H,d}\left(\frac{b_h}{a_h}\right)^{-\frac{1}{2}} = \frac{\epsilon}{2}\Big(\sum_{t=0}^{H-1}((1-\gamma^{t+1})\gamma^{H-t-1})^{\frac{1}{2}}\Big)^{-1}\left(\frac{\gamma^{H-h-1}}{(1-\gamma^{h+1})}\right)^{-\frac{1}{2}},$$

is the optimal solution to this problem. For $H = \left\lceil\frac{\log(\frac{(1-\gamma)\epsilon}{2R^*})}{\log(\gamma)}\right\rceil$ we have that $\frac{\gamma^H R^*}{1-\gamma} \leq \frac{\epsilon}{2}$. Using these $(\epsilon_h)_{h=0}^{H-1}$ in Theorem 4.6 results in a value function error for DynPG bounded by

$$\|V_\infty^* - V_H^{\hat{\pi}_H^*}\|_\infty \leq \frac{\gamma^H R^*}{1-\gamma} + \sum_{h=0}^{H-1}\gamma^{H-h-1}\epsilon_h \leq \frac{\epsilon}{2} + \frac{\epsilon}{2} = \epsilon. \tag{17}$$

For the optimal complexity bound we ignore the Gauss-brackets in $N_h$ and derive

$$\sum_{h=0}^{H-1} N_h = \frac{4R^*|\mathcal{A}|^2}{(1-\gamma)} \Big\| \frac{1}{\mu} \Big\|_\infty \sum_{h=0}^{H-1} a_h c_h^{-1}$$

$$= \frac{4R^*|\mathcal{A}|^2}{(1-\gamma)} \Big\| \frac{1}{\mu} \Big\|_\infty C_{H,d}^{-1} \sum_{h=0}^{H-1} a_h \Big(\frac{b_h}{a_h}\Big)^{\frac{1}{2}}$$

$$= \frac{4R^*|\mathcal{A}|^2}{(1-\gamma)} \Big\| \frac{1}{\mu} \Big\|_\infty C_{H,d}^{-1} \sum_{h=0}^{H-1} (a_h b_h)^{\frac{1}{2}}$$

$$= \frac{4R^*|\mathcal{A}|^2}{(1-\gamma)} \Big\| \frac{1}{\mu} \Big\|_\infty \frac{1}{d} \Big(\sum_{h=0}^{H-1} (a_h b_h)^{\frac{1}{2}}\Big)^2$$

$$= \frac{8R^*|\mathcal{A}|^2}{(1-\gamma)\epsilon} \Big\| \frac{1}{\mu} \Big\|_\infty \Big(\sum_{h=0}^{H-1} ((1-\gamma^{h+1})\gamma^{H-h-1})^{\frac{1}{2}}\Big)^2$$

$$= \frac{8R^*|\mathcal{A}|^2}{(1-\gamma)\epsilon} \Big\| \frac{1}{\mu} \Big\|_\infty \Big(\sum_{h=0}^{H-1} ((\gamma^{H-h-1} - \gamma^H))^{\frac{1}{2}}\Big)^2$$

$$\leq \frac{8R^*|\mathcal{A}|^2}{(1-\gamma)\epsilon} \Big\| \frac{1}{\mu} \Big\|_\infty H \sum_{h=0}^{H-1} (\gamma^{H-h-1} - \gamma^H),$$

where we used Jensen's inequality in the last step. Further it holds that

$$\sum_{h=0}^{H-1} N_h \leq \frac{8R^*|\mathcal{A}|^2}{(1-\gamma)\epsilon} \Big\| \frac{1}{\mu} \Big\|_\infty H \sum_{h=0}^{H-1} (\gamma^{H-h-1} - \gamma^H)$$

$$\leq \frac{8R^*|\mathcal{A}|^2}{(1-\gamma)\epsilon} \Big\| \frac{1}{\mu} \Big\|_\infty H \sum_{h=0}^{H-1} \gamma^h$$

$$\leq \frac{8R^*|\mathcal{A}|^2}{(1-\gamma)\epsilon} \Big\| \frac{1}{\mu} \Big\|_\infty H \frac{1}{1-\gamma}$$

$$= \frac{8R^* H |\mathcal{A}|^2}{(1-\gamma)^2 \epsilon} \Big\| \frac{1}{\mu} \Big\|_\infty.$$

Finally, note that we can rewrite $H$ to be

$$H = \Big\lceil \log_\gamma \Big(\frac{(1-\gamma)\epsilon}{2R^*}\Big) \Big\rceil = \Big\lceil \frac{\log\Big(\frac{(1-\gamma)\epsilon}{2R^*}\Big)}{\log(\gamma)} \Big\rceil = \Big\lceil \frac{\log((1-\gamma)^{-1}\epsilon^{-1} 2R^*)}{\log(\gamma^{-1})} \Big\rceil$$

$\square$

**Overall error.** To deal with the overall error, we first derive the following upper bound.

**Lemma D.7.** *Assume zero approximation error, i.e. $T^* = \sup_\theta T^{\pi_\theta}$. Further, assume bounded optimization errors $\|T^*(V_h^{\hat{\pi}_h^*}) - T^{\hat{\pi}_h^*}(V_h^{\hat{\pi}_h^*})\|_\infty \leq \epsilon_h$ for all $h = 0,\ldots,H$. If $\epsilon_H \leq (1-\gamma)\sum_{h=0}^{H-1}\gamma^{H-h-1}\epsilon_h$, then the overall error of DynPG is bounded by*

$$\|V_\infty^* - V_\infty^{\hat{\pi}_H^*}\|_\infty \leq \frac{3}{1-\gamma}\Big(\frac{\gamma^H R^*}{1-\gamma} + \sum_{h=0}^{H-1}\gamma^{H-h-1}\epsilon_h\Big).$$

*Proof.* First, we have by triangle inequality

$$\|V_{H+1}^{\hat{\pi}_{H+1}^*} - V_H^{\hat{\pi}_H^*}\|_\infty \leq \|V_{H+1}^{\hat{\pi}_{H+1}^*} - V_\infty^*\|_\infty + \|V_\infty^* - V_H^{\hat{\pi}_H^*}\|_\infty.$$

By Proposition 4.1 we obtain, under zero approximation error, that

$$\|V_{H+1}^{\hat{\pi}^*_{H+1}} - V_\infty^*\|_\infty + \|V_\infty^* - V_H^{\hat{\pi}^*_H}\|_\infty \leq \frac{\gamma^{H+1} R^*}{1-\gamma} + \sum_{h=0}^{H} \gamma^{H-h} \epsilon_h + \frac{\gamma^H R^*}{1-\gamma} + \sum_{h=0}^{H-1} \gamma^{H-h-1} \epsilon_h.$$

By the assumption $\epsilon_H \leq (1-\gamma) \sum_{h=0}^{H-1} \gamma^{H-h-1} \epsilon_h$, it holds further that

$$\sum_{h=0}^{H} \gamma^{H-h} \epsilon_h \leq \gamma \sum_{h=0}^{H-1} \gamma^{H-h-1} \epsilon_h + \epsilon_H \leq \sum_{h=0}^{H-1} \gamma^{H-h-1} \epsilon_h.$$

We obtain

$$\|V_{H+1}^{\hat{\pi}^*_{H+1}} - V_H^{\hat{\pi}^*_H}\|_\infty \leq \frac{2\gamma^H R^*}{1-\gamma} + 2 \sum_{h=0}^{H-1} \gamma^{H-h-1} \epsilon_h.$$

We deduce for the overall error (by Proposition 4.1 and under zero approximation error) that

$$\|V_\infty^* - V_\infty^{\hat{\pi}^*_H}\|_\infty \leq \frac{\gamma^H R^*}{1-\gamma} + \sum_{h=0}^{H-1} \gamma^{H-h-1} \epsilon_h + \frac{1}{1-\gamma}\left(\frac{2\gamma^H R^*}{1-\gamma} + 2 \sum_{h=0}^{H-1} \gamma^{H-h-1} \epsilon_h\right)$$

$$\leq \frac{3}{1-\gamma}\left(\frac{\gamma^H R^*}{1-\gamma} + \sum_{h=0}^{H-1} \gamma^{H-h-1} \epsilon_h\right).$$

$\square$

It is important to notice that the overall error is upper bounded by the same error terms, $\frac{\gamma^H R^*}{1-\gamma} + \sum_{h=0}^{H-1} \gamma^{H-h-1} \epsilon_h$, as the value function error (under zero approximation error) up to the constant $\frac{3}{1-\gamma}$. Thus, we can obtain the result for the overall error by substituting $\epsilon$ with $\frac{(1-\gamma)\epsilon}{3}$.

**Theorem D.8.** *[Detailed version of Theorem 4.8 (1.)]*
*Let Assumption 4.5 hold true and the gradient be accessed exactly. Choose $\epsilon > 0$ and set*

$$H = \lceil \log_\gamma \left(\frac{(1-\gamma)^2 \epsilon}{6R^*}\right)\rceil,$$

$$\epsilon_h = \frac{\epsilon(1-\gamma)}{6}\left(\sum_{h=0}^{H-1}(((1-\gamma^{h+1})\gamma^{H-h-1})^{\frac{1}{2}})\right)^{-1}\left(\frac{\gamma^{H-h-1}}{(1-\gamma^{h+1})}\right)^{-\frac{1}{2}}, \quad \forall h = 0\ldots, H-1$$

$$\epsilon_H = \frac{(1-\gamma)\epsilon}{6}$$

$$\eta_h = \frac{1-\gamma}{2R^*(1-\gamma^{h+1})}, \quad \forall h = 0\ldots, H$$

$$N_h = \left\lceil \frac{4R^*(1-\gamma^{h+1})|\mathcal{A}|^2}{(1-\gamma)\epsilon_h}\left\|\frac{1}{\mu}\right\|_\infty\right\rceil, \quad \forall h = 0\ldots, H.$$

*Then, the stationary policy $\hat{\pi}^*_H$ obtained by DynPG under exact gradients achieves $\|V_\infty^* - V_\infty^{\hat{\pi}^*_H}\|_\infty \leq \epsilon$. The total number of gradient steps are given by*

$$\sum_{h=0}^{H} N_h = \frac{48R^*|\mathcal{A}|^2}{(1-\gamma)^3 \epsilon}\left\lceil \frac{\log(6R^*(1-\gamma)^{-2}\epsilon^{-1})}{\log(\gamma^{-1})}\right\rceil\left\|\frac{1}{\mu}\right\|_\infty.$$

*Proof.* The optimization procedure is the same as for the value function error by substituting $\epsilon$ with $\frac{(1-\gamma)\epsilon}{3}$. Moreover, note that

$$\sum_{h=0}^{H-1} \gamma^{H-h-1} \epsilon_h = \frac{(1-\gamma)\epsilon}{6}.$$

Thus, $\epsilon_H \leq \frac{(1-\gamma)\epsilon}{6}$ is sufficient for Lemma D.7 to hold, and we obtain that using these $(\epsilon_h)$'s in Theorem 4.6 results in an overall error for DynPG given by

$$\|V_\infty^* - V_\infty^{\hat{\pi}_H^*}\|_\infty \leq \frac{3}{1-\gamma}\Big(\frac{\gamma^H R^*}{1-\gamma} + \sum_{h=0}^{H-1}\gamma^{H-h-1}\epsilon_h\Big) \leq \frac{3}{1-\gamma}\frac{\epsilon(1-\gamma)}{3} = \epsilon.$$

Note that we can again rewrite $H$ to be

$$H = \Big\lceil \log_\gamma\Big(\frac{(1-\gamma)^2\epsilon}{6R^*}\Big)\Big\rceil = \Big\lceil \frac{\log((1-\gamma)^{-2}\epsilon^{-1}6R^*)}{\log(\gamma^{-1})}\Big\rceil.$$

Thus, the complexity bounds for the optimization epochs $h = 0, \ldots, H$ are given by

$$\sum_{h=0}^H N_h = \sum_{h=0}^{H-1} N_h + N_H$$
$$= \frac{24R^*|\mathcal{A}|^2}{(1-\gamma)^3\epsilon}\Big\lceil\frac{\log((1-\gamma)^{-2}\epsilon^{-1}6R^*)}{\log(\gamma^{-1})}\Big\rceil\Big\|\frac{1}{\mu}\Big\|_\infty + \frac{24R^*(1-\gamma^{H+1})|\mathcal{A}|^2}{(1-\gamma)^2\epsilon}\Big\|\frac{1}{\mu}\Big\|_\infty$$
$$\leq \frac{48R^*|\mathcal{A}|^2}{(1-\gamma)^3\epsilon}\Big\lceil\frac{\log((1-\gamma)^{-2}\epsilon^{-1}6R^*)}{\log(\gamma^{-1})}\Big\rceil\Big\|\frac{1}{\mu}\Big\|_\infty,$$

where we used the calculations in the proof of Theorem D.6 in the first step and substituted $\epsilon$ by $\frac{(1-\gamma)\epsilon}{3}$. □

**Lemma D.9.** *The term* $\log(\gamma^{-1}) = -\log(\gamma)$ *is asymptotically equivalent to* $(1-\gamma)$ *for* $\gamma \uparrow 1$.

*Proof.* By the definition of asymptotic equivalence for two functions $f$ and $g$ we have to show that

$$\lim_{x\uparrow 1}\frac{f(x)}{g(x)} = 1.$$

It holds by L'Hospital rule that

$$\lim_{\gamma\uparrow 1}\frac{-\log(\gamma)}{(1-\gamma)} = \lim_{\gamma\uparrow 1}\frac{-\gamma^{-1}}{-1} = 1.$$

□

# E DynPG in Large or Complex Environments

In the following, we aim to discuss a challenge that may arise when applying DynPG in practice. In very complex MDPs, where the policy parametrization needs to be rich, the DynPG approach in Algorithm 1 can suffer from a storage problem. The number of policy parameters that need to be stored for future decisions scale linearly with the number of training steps. In order to circumvent this problem for practical applications we propose an actor-critic based modification of DynPG, called Dynamic Actor-Critic (DynAC). The general idea behind actor-critic in vanilla PG is to introduce a so called critic, a second parametrized class of functions $(Q^w)_{w\in\mathbb{R}^l}$, which approximates the Q-function $Q^{\pi^\theta}$ in the policy gradient theorem. Using the critic as estimator of the Q-values no more roll-outs are needed to estimate the rewards-to-go. In the actor-critic variant of DynPG, we include an additional training procedure after optimizing policy $\hat{\pi}_h^*$ to update the critic based on the old critic, $Q^{w_h} \approx \mathcal{Q}_h^{\hat{\pi}_{h-1}^*,\ldots,\hat{\pi}_0^*}$, and the newly trained policy, $\hat{\pi}_h^*$, using the relation:

$$Q_{h+1}^{\hat{\pi}_h^*,\ldots,\hat{\pi}_0^*}(s,a) = r(s,a) + \gamma\sum_{s'}p(s'|s,a)\sum_{a'}\hat{\pi}_h^*(a'|s')\mathcal{Q}_h^{\hat{\pi}_{h-1}^*,\ldots,\hat{\pi}_0^*}.$$

The critic is used in the policy gradient theorem as an estimator for $Q_{h+1}^{\pi_h}$ such that the new gradient in DynAC is given by

$$G^{\text{Dyn-AC}} = \nabla_\theta\log(\pi^\theta(A|S))Q^{w_{h+1}}(S,A),$$

---

**Algorithm 2:** DynAC

---

**Input:** Initial state distribution $\mu$, class of policies $(\pi^\theta)_{\theta \in \mathbb{R}^d}$, class of critic parametrization
   $(Q_w)_{w \in \mathbb{R}^l}$.
**Result:** Approximation of $\pi^*$, denoted as $\hat{\pi}^*$.

1  Set $h = 0$;
2  Train $Q^{w_1}$ to approximate the reward function $r$;
3  **while** *Convergence criterion not met* **do**
4       Initialize $\theta_0$ (e.g., $\theta_0 \equiv 0$);
5       Choose $\alpha_h$ and $N_h$ (cf., Remark 4.4);
6       **for** $n = 0, \ldots, N_h - 1$ **do**
7           Sample $S \sim \mu$ and $A \sim \pi^\theta(\cdot|S)$;
8           Set $G^{\text{Dyn-AC}} = \nabla_\theta \log(\pi^\theta(A|S)) Q^{w_{h+1}}(S, A)$;
9           Update $\theta_{n+1} = \theta_n + \alpha_h G$;
10      **end**
11      Set $\hat{\pi}_h^* = \pi^{\theta_{N_h}}$;
12      Train $w_{h+2}$ s.t. for all $s, a$: $Q^{w_{h+2}}(s, a) \approx \mathbb{E}_{S \sim p(\cdot|s,a), A \sim \hat{\pi}_h^*(\cdot|S)}[r(s, a) + \gamma Q^{w_{h+1}}(S, A)]$ ;
13      Set $h = h + 1$;
14 **end**
15 Return $\hat{\pi}^* = \hat{\pi}_{h-1}^*$;

---

where $S \sim \mu$ and $A \sim \pi^\theta(\cdot|s)$ and $\pi^\theta$ is the policy we are currently training for epoch $h + 1$. This way there is no need to store all trained policies only the currently trained policy and the current critic are needed. For applications where the performance of vanilla PG is poor and the parametrized policy needs to be very rich such that storing many policies might cause storage issues, we suggest to keep this variant of DynPG in mind. We call this approach DynAC for dynamic actor-critic and summarize the steps in Algorithm 2.

A theoretical analysis of this approach would require an assumption on the approximation error to train the Q-functions. As the gradients are no longer unbiased, convergence towards the global optimum cannot be guaranteed and the bias errors will appear in the overall error. We leave further investigations of DynAC to future work, as an in-depth exploration would exceed the scope of this paper.

## F   Convergence Rates under Tabular Softmax Parametrization - Sampled Gradients

In this section we drop the exact gradient assumption and analyze DynPG for sample based gradient estimates (generated by MDP roll-outs). We emphasize that the main insight into DynPG (breaking the (possibly) exponential $\gamma$-dependence) is part of the exact gradient analysis. Even though the estimates provided here are not relevant for practical use, we add this discussion for completeness.

For a fixed future policy $\hat{\pi}_h^*$, consider $K_h$ trajectories $(s_k^i, a_k^i)_{k=0}^h$, for $i = 1, \ldots, K_h$, generated by $s_0^i \sim \mu$, $a_0^i \sim \pi_\theta$ and $a_k^i \sim \hat{\pi}_{h-k}^*$ for $1 \leq k \leq h$. In analogy to the gradient estimator for vanilla PG (motivated by the policy gradient theorem) the estimator of $\nabla_\theta V_{h+1}^{\pi_\theta, \hat{\pi}_h^*}(\mu)$ is defined by

$$\widehat{G}_{K_h}(\theta) = \frac{1}{K_h} \sum_{i=1}^{K_h} \nabla \log(\pi_\theta(a_0^i|s_0^i)) \hat{Q}_{h+1}^i, \tag{18}$$

where $\hat{Q}_{h+1}^i = \sum_{k=0}^h \gamma^k r(s_k^i, a_k^i)$ is an unbiased estimator of $Q_{h+1}^{\hat{\pi}_h^*}(s_0^i, a_0^i)$. Then the stochastic PG update for training the parameter $\theta$ in epoch $h$ is given by

$$\bar{\theta}_{n+1} = \bar{\theta}_n + \eta_h \widehat{G}_{K_h}(\bar{\theta}_n). \tag{19}$$

Our main result for the optimization of the contextual bandits in DynPG is given as follows.

**Theorem F.1.** *Let Assumption 4.5 hold true and DynPG be used with stochastic updates from* (19). *Suppose, we are in epoch $h$ with fixed future policy $\Lambda = \hat{\pi}_h^*$ and for any $\delta_h > 0$ and $\epsilon_h > 0$, choose*

$$N_h \geq \left(\frac{12|\mathcal{A}|^2(1-\gamma^{h+1})R^*}{(1-\gamma)\epsilon\delta}\right)^2, \; \eta_h = \frac{1-\gamma}{2(1-\gamma^{h+1})R^*\sqrt{N_h}} \text{ and } K_h \geq \frac{5|\mathcal{A}|^2 N_h^3}{\delta^2}. \text{ Then, it holds}$$

$$\mathbb{P}\Big(T^*(V_h^\Lambda)(\mu) - V_{h+1}^{\{\pi_{\bar{\theta}_n},\Lambda\}}(\mu) \geq \epsilon_h\Big) \leq \delta_h.$$

To prove Theorem F.1 we adapt the proof ideas in [12, Sec. D.2] for finite-time dynamic policy gradient to our discounted setting. We restate all results and show how to adapt the proofs.

We first verify that the gradient estimator in Appendix F is unbiased and has bounded variance.

**Lemma F.2.** *Consider the estimator $\hat{G}_{K_h}(\theta)$ from Appendix F. For any $K_h > 0$ it holds that*

$$\mathbb{E}_\mu^{\{\pi_\theta,\Lambda\}}[\hat{G}_{K_h}(\theta)] = \nabla_\theta V_{h+1}^{\{\pi_\theta,\Lambda\}}(\mu)$$

*and under softmax parametrization*

$$\mathbb{E}_\mu^{\{\pi_\theta,\Lambda\}}[\|\hat{G}_{K_h}(\theta) - \nabla_\theta V_{h+1}^{\{\pi_\theta,\Lambda\}}(\mu)\|^2] \leq \frac{5(\frac{1-\gamma^{h+1}}{1-\gamma}R^*)^2}{K_h} =: \frac{\xi_h}{K_h}$$

*Proof.* By the definition of $\hat{G}_{K_h}(\theta)$ we have

$$\mathbb{E}_\mu^{\{\pi_\theta,\Lambda\}}[\hat{G}_{K_h}(\theta)] = \mathbb{E}_\mu^{\{\pi_\theta,\Lambda\}}\Big[\nabla\log(\pi_\theta(A_0|S_0))\sum_{k=0}^h \gamma^k r(S_k, A_k)\Big],$$

as in [12, Lem. D.6] and from the DynPG policy gradient theorem (Theorem C.3), we obtain the unbiasedness directly.

For the second claim, we first obtain that

$$\mathbb{E}_\mu^{\{\pi_\theta,\Lambda\}}\Big[\|\hat{G}_{K_h}(\theta) - \nabla_\theta V_{h+1}^{\{\pi_\theta,\Lambda\}}(\mu)\|^2\Big]$$

$$\leq \frac{1}{K_h}\mathbb{E}_\mu^{\{\pi_\theta,\Lambda\}}\Big[\|\nabla\log(\pi_\theta(A_0|S_0))\hat{Q}_{h+1}(S_0, A_0) - \nabla_\theta V_{h+1}^{\{\pi_\theta,\Lambda\}}(\mu)\|^2\Big]$$

$$= \frac{1}{K_h}\mathbb{E}_\mu^{\{\pi_\theta,\Lambda\}}\Big[\sum_{s\in\mathcal{S}}\sum_{a\in\mathcal{A}}\Big(\mathbf{1}_{s=S_0}(\mathbf{1}_{a=A_0} - \pi_\theta(a|s))\sum_{k=0}^h \gamma^k r(S_k, A_k) - \mu(s)\pi_\theta(a|s)Q_{h+1}^\Lambda(s,a)\Big)^2\Big].$$

Note that we have nearly the same term as in the proof of [12, Lem. D.6]. We can follow line by line the arguments there and have to replace the bounds $\sum_{k=h}^{H-1} r(S_k, A_k) \leq (H-h)R^*$ with our upper bound $\sum_{k=0}^h \gamma^k r(S_k, A_k) \leq \frac{1-\gamma^{h+1}}{1-\gamma}R^*$.

So replacing $(H-h)$ with $\frac{1-\gamma^{h+1}}{1-\gamma}$, we obtain that

$$\mathbb{E}_\mu^{\{\pi_\theta,\Lambda\}}\Big[\|\hat{G}_{K_h}(\theta) - \nabla_\theta V_{h+1}^{\{\pi_\theta,\Lambda\}}(\mu)\|^2\Big] \leq \frac{5(\frac{1-\gamma^{h+1}}{1-\gamma}R^*)^2}{K_h}.$$

$\square$

We continue to follow the proof in [12, Sec. D.2] and introduce a stopping time to control the non-uniform gradient domination inequality along the exact gradient trajectories.

Recall that we consider the optimization of DynPG at epoch $h$. Let $(\bar{\theta}_n)_{n\geq 0}$ be the stochastic process from (19) and let $(\theta_n)_{n\geq 0}$ be the deterministic sequence generated by DynPG with exact gradients. Assume in the following that the initial parameter agrees, i.e. $\theta_0 = \bar{\theta}_0$, and the step size $\eta_h$ is the same for both processes.

We denote by $(\mathcal{F}_n)_{n\geq 0}$ the natural filtration of $(\bar{\theta}_n)_{n\geq 0}$. As we assume uniform initialization, the non-uniform gradient domination property holds along the deterministic gradient scheme with constant $c_h = \min_{n\geq 0}\min_{s\in\mathcal{S}}\pi^{\theta_n}(a^*(s)|s) = \frac{1}{|\mathcal{A}|}$ (Lemma D.4).

Therefore we define the stopping time

$$\tau := \min\{n \geq 0 : \|\theta_n - \bar{\theta}_n\|_2 \geq \frac{1}{4|\mathcal{A}|}\}.$$

Employing this stopping time, we can control the non-uniform gradient domination property along the stochastic gradient trajectory, before the stopping time is hit.

**Lemma F.3.** *Let Assumption 4.5 hold true and DynPG be used with stochastic updates from* (19). *Suppose, we are in epoch $h$ with fixed future policy $\Lambda = \hat{\pi}_h^*$. Then, it holds almost surely that $\min_{0 \leq n \leq \tau} \min_{s \in \mathcal{S}} \pi_{\bar{\theta}_n}(a^*(s)|s) \geq \frac{1}{2|\mathcal{A}|}$ is strictly positive.*

*Proof.* The proof follows line by line as in [12, Lem. D.7] with $c_h = \frac{1}{|\mathcal{A}|}$. □

We consider the two events $\{n \leq \tau\}$ and $\{n \geq \tau\}$ separately. On the event $\{n \leq \tau_h\}$ we obtain the following convergence rate.

**Lemma F.4.** *Let Assumption 4.5 hold true and DynPG be used with stochastic updates from* (19). *Suppose, we are in epoch $h$ with fixed future policy $\Lambda = \hat{\pi}_h^*$. Suppose, that*

- *the batch size $(K_h)_n \geq \frac{45}{64|\mathcal{A}|^2 N_h^{\frac{3}{2}}}(1 - \frac{1}{2\sqrt{N_h}})n^2$ is increasing for some $N_h \geq 1$*

- *the step size $\eta_h = \frac{1-\gamma}{2(1-\gamma^{h+1})R^*\sqrt{N_h}}$.*

*Then,*

$$
\mathbb{E}_\mu^{\{\pi_\theta, \Lambda\}}\left[(\sup_\theta V_{h+1}^{\{\pi_\theta, \Lambda\}}(\mu) - V_{h+1}^{\{\pi_{\bar{\theta}_n}, \Lambda\}}(\mu))\mathbf{1}_{\{n \leq \tau\}}\right] \leq \frac{32\sqrt{N_h}(1-\gamma^{h+1})|\mathcal{A}|^2 R^*}{3(1-\gamma)(1 - \frac{1}{2\sqrt{N_h}})n}.
$$

*Proof.* Line by line as in [12, Lem. D.8]. Note that we have a different smoothness constant and a different variance control. In both constants one has to replace $(H - h)$ in [12, Lem. D.8] with $\frac{1-\gamma^{h+1}}{1-\gamma}$ and $c_h$ with $\frac{1}{|\mathcal{A}|}$. □

Next, we bound the probability of the event $\{n \geq \tau\}$ by $\delta$ for a large enough batch size $K_h$.

**Lemma F.5.** *Let Assumption 4.5 hold true and DynPG be used with stochastic updates from* (19). *Suppose, we are in epoch $h$ with fixed future policy $\Lambda = \hat{\pi}_h^*$. For any $\delta > 0$, suppose that*

- *the batch size $K_h \geq \frac{5|\mathcal{A}|^2 n^3}{\delta^2}$*

- *the step size $\eta_h \leq \frac{1}{\sqrt{n}L_h}$.*

*Then, we have $\mathbb{P}(n \geq \tau) < \delta$.*

*Proof.* Line by line as in [12, Lem. D.9]. Replace again $c_h$ with $\frac{1}{|\mathcal{A}|}$ and $(H - h)$ with $\frac{1-\gamma^{h+1}}{1-\gamma}$. □

Using the above Lemmas we can prove the convergence of each contextual bandit problem.

*Proof of Theorem F.1.* Proof works as in [12, Lem. D.10], we can separate the probability using the stopping time and under the zero approximation error of softmax we have that

$$
\mathbb{P}\Big(T^*(V_h^\Lambda)(\mu) - V_{h+1}^{\{\pi_{\bar\theta_n},\Lambda\}}(\mu) \geq \epsilon_h\Big)
$$

$$
= \mathbb{P}\Big(\sup_\theta V_{h+1}^{\{\pi_\theta,\Lambda\}}(\mu) - V_{h+1}^{\{\pi_{\bar\theta_{N_h}},\Lambda\}}(\mu) \geq \epsilon\Big)
$$

$$
\leq \mathbb{P}\Big(\{N_h \leq \tau\} \cap \{\sup_\theta V_{h+1}^{\{\pi_\theta,\Lambda\}}(\mu) - V_{h+1}^{\{\pi_{\bar\theta_{N_h}},\Lambda\}}(\mu) \geq \epsilon\}\Big)
$$

$$
\quad + \mathbb{P}\Big(\{N_h \geq \tau\} \cap \{\sup_\theta V_{h+1}^{\{\pi_\theta,\Lambda\}}(\mu) - V_{h+1}^{\{\pi_{\bar\theta_{N_h}},\Lambda\}}(\mu) \geq \epsilon\}\Big)
$$

$$
\leq \frac{\mathbb{E}\Big[(\sup_\theta V_{h+1}^{\{\pi_\theta,\Lambda\}}(\mu) - V_{h+1}^{\{\pi_{\bar\theta_{N_h}},\Lambda\}}(\mu))\mathbf{1}_{\{\tau_h \geq N_h\}}\Big]}{\epsilon} + \mathbb{P}(N_h \geq \tau)
$$

$$
\leq \frac{1}{\epsilon}\frac{32\sqrt{N_h}|\mathcal{A}|^2(1-\gamma^{h+1})R^*}{3(1-\gamma)(1-\frac{1}{2\sqrt{N_h}})N_h} + \frac{\delta}{2}
$$

$$
\leq \frac{\delta}{2} + \frac{\delta}{2}
$$

$$
= \delta,
$$

where the second inequality is due to Lemma F.4 and Lemma F.5. The last inequality follows by our choice of $N_h$ as in [12, Lem. D.10] by replacing $c_h$ with $\frac{1}{|\mathcal{A}|}$ and $(H-h)$ with $\frac{1-\gamma^{h+1}}{1-\gamma}$. $\qquad\square$

From the previous contextual bandit result, we can derive a sample complexity result under which we obtain convergence in stochastic DynPG with arbitrary high probability.

**Theorem F.6.** *Let Assumption 4.5 hold true and DynPG be used with stochastic updates from* (19)*. Choose $\epsilon > 0$, $\delta > 0$ and set*

$$
H = \lceil \log_\gamma \big(\frac{(1-\gamma)^2\epsilon}{6R^*}\big)\rceil,
$$

$$
\epsilon_h = \frac{\epsilon(1-\gamma)}{6}\Big(\sum_{h=0}^{H-1}(((1-\gamma^{h+1})\gamma^{H-h-1})^{\frac{1}{2}})\Big)^{-1}\Big(\frac{\gamma^{H-h-1}}{(1-\gamma^{h+1})}\Big)^{-\frac{1}{2}}, \quad \forall h = 0\ldots, H-1,
$$

$$
\epsilon_H = \frac{(1-\gamma)\epsilon}{6},
$$

$$
N_h = \Big\lceil\big(\frac{12|\mathcal{A}|^2(H+1)(1-\gamma^{h+1})R^*}{(1-\gamma)\epsilon_h\delta}\big)^2\big\|\frac{1}{\mu}\big\|_\infty^2\Big\rceil, \quad \forall h = 0\ldots, H,
$$

$$
\eta_h = \frac{1-\gamma}{2(1-\gamma^{h+1})R^*\sqrt{N_h}}, \quad \forall h = 0\ldots, H,
$$

$$
K_h = \Big\lceil\frac{5|\mathcal{A}|^2(H+1)^2N_h^3}{\delta^2}\Big\rceil, \quad \forall h = 0\ldots, H.
$$

*Then, the stationary policy $\hat\pi_H^*$ obtained by stochastic DynPG achieves $\mathbb{P}(\|V_\infty^* - V_\infty^{\hat\pi_H^*}\|_\infty < \epsilon) \geq 1 - \delta$. The total number of samples is bounded by*

$$
\sum_{h=0}^H N_h K_h \leq \Big\|\frac{1}{\mu}\Big\|_\infty^8\frac{c_2|\mathcal{A}|^{18}(R^*)^8}{(1-\gamma)^{17}\delta^{10}\epsilon^8}\Big\lceil\frac{\log((1-\gamma)^{-1}\epsilon^{-1}2R^*)}{\log(\gamma^{-1})}\Big\rceil^{18}.
$$

Note first, that the constant $c_2$ in the sample complexity is indeed a natural number, hence independent of any model parameters. The concrete values of $H$, $(N_h)_{h=0}^H$, $(\eta_h)_{h=0}^H$ and $(K_h)_{h=0}^H$ are derived in the proof below.

**Remark F.7.** *We obtain again a sample complexity that scales at most polynomial in $(1-\gamma)^{-1}$ and guarantees convergence in high probability. In [7] a sample complexity for softmax PG with entropy regularization is derived, where even larger batch sizes are required, which leads to an exponentially*

*bad sample complexity with respect to $(1 - \gamma)^{-1}$. Still, we do not expect the rates in Theorem F.6 to be tight, but they prove that convergence in stochastic settings is achievable. We also want to remind the reader of the sample based example presented in Line 13, where no batch is needed for the implementation.*

*Proof.* By our choice of $N_h$, $\eta_h$ and $K_h$ and Theorem F.1, it holds that

$$\mathbb{P}\Big(\|\sup_\theta V_{h+1}^{\{\pi_\theta, \Lambda\}} - V_{h+1}^{\{\pi_{\bar\theta_{N_h}}, \Lambda\}}\|_\infty < \epsilon_h\Big) = 1 - \mathbb{P}\Big(\exists s \in \mathcal{S} : \sup_\theta V_{h+1}^{\{\pi_\theta, \Lambda\}}(s) - V_{h+1}^{\{\pi_{\bar\theta_{N_h}}, \Lambda\}}(s) \geq \epsilon_h\Big)$$

$$\geq 1 - \mathbb{P}\Big(\sup_\theta V_{h+1}^{\{\pi_\theta, \Lambda\}}(\mu) - V_{h+1}^{\{\pi_{\bar\theta_{N_h}}, \Lambda\}}(\mu) \geq \epsilon_h \Big\|\frac{1}{\mu}\Big\|_\infty^{-1}\Big)$$

$$\geq 1 - \frac{\delta}{H},$$

where in the first inequality we have used we used that

$$\sup_\theta V_{h+1}^{\{\pi_\theta, \Lambda\}}(s) - V_{h+1}^{\{\pi_{\bar\theta_{N_h}}, \Lambda\}}(s) \leq \Big\|\frac{1}{\mu}\Big\|_\infty \sup_\theta V_{h+1}^{\{\pi_\theta, \Lambda\}}(\mu) - V_{h+1}^{\{\pi_{\bar\theta_{N_h}}, \Lambda\}}(\mu).$$

Next we introduce the event $C = \Big\{\forall h = 0, \ldots, H : \|\sup_\theta V_{h+1}^{\{\pi_\theta, \Lambda\}} - V_{h+1}^{\{\pi_{\bar\theta_{N_h}}, \Lambda\}}\|_\infty < \epsilon_h\Big\}$. By Lemma D.7 we deduce that under the event $C$ it holds almost surely that

$$\|V_\infty^* - V_\infty^{\hat\pi_H^*}\|_\infty < \frac{3}{1-\gamma}\Big(\frac{\gamma^H R^*}{1-\gamma} + \sum_{h=0}^{H-1} \gamma^{H-h-1} \epsilon_h\Big) \leq \epsilon$$

by our choices of $\epsilon_h$'s and $H$. So $C \subseteq \{\|V_\infty^* - V_\infty^{\hat\pi_H^*}\|_\infty < \epsilon\}$ and therefore,

$$\mathbb{P}\Big(\|V_\infty^* - V_\infty^{\hat\pi_H^*}\|_\infty < \epsilon\Big)$$

$$\geq \mathbb{P}\Big(\forall h = 0, \ldots, H : \|\sup_\theta V_{h+1}^{\{\pi_\theta, \Lambda\}} - V_{h+1}^{\{\pi_{\bar\theta_{N_h}}, \Lambda\}}\|_\infty < \epsilon_h\Big)$$

$$= 1 - \mathbb{P}\Big(\exists h = 0, \ldots, H : \|\sup_\theta V_{h+1}^{\{\pi_\theta, \Lambda\}} - V_{h+1}^{\{\pi_{\bar\theta_{N_h}}, \Lambda\}}\|_\infty \geq \epsilon_h\Big)$$

$$\geq 1 - \sum_{h=0}^H \mathbb{P}\Big(\|\sup_\theta V_{h+1}^{\{\pi_\theta, \Lambda\}} - V_{h+1}^{\{\pi_{\bar\theta_{N_h}}, \Lambda\}}\|_\infty \geq \epsilon_h\Big)$$

$$\geq 1 - \sum_{h=0}^H \frac{\delta}{H+1}$$

$$= 1 - \delta.$$

To upper bound the total number of samples, we suppress the Gauss-brackets for simplicity and compute

$$\sum_{h=0}^{H} N_h K_h$$

$$=\sum_{h=0}^{H} \frac{5|\mathcal{A}|^2(H+1)^2 N_h^4}{\delta^2}$$

$$=\sum_{h=0}^{H} \frac{5|\mathcal{A}|^2(H+1)^2 N_h^4}{\delta^2}$$

$$=\sum_{h=0}^{H} \frac{c_1|\mathcal{A}|^{18}(H+1)^{10}(1-\gamma^{h+1})^8(R^*)^8}{(1-\gamma)^8\epsilon_h^8\delta^{10}}\Big\|\frac{1}{\mu}\Big\|_\infty^8$$

$$=\Big\|\frac{1}{\mu}\Big\|_\infty^8 \frac{c_1|\mathcal{A}|^{18}(H+1)^{10}(R^*)^8}{(1-\gamma)^8\delta^{10}}\sum_{h=0}^{H} \frac{(1-\gamma^{h+1})^8}{\epsilon_h^8}$$

$$=\Big\|\frac{1}{\mu}\Big\|_\infty^8 \frac{c_2|\mathcal{A}|^{18}(H+1)^{10}(R^*)^8}{(1-\gamma)^{16}\delta^{10}\epsilon^8}\Big(\sum_{h=0}^{H-1}(((1-\gamma^{h+1})\gamma^{H-h-1})^{\frac{1}{2}})\Big)^8 \sum_{h=0}^{H-1}(1-\gamma^{h+1})^4(\gamma^{H-h-1})^4$$

$$+\Big\|\frac{1}{\mu}\Big\|_\infty^8 \frac{c_2|\mathcal{A}|^{18}(H+1)^{10}(R^*)^8}{(1-\gamma)^8\delta^{10}}\frac{(1-\gamma^{H+1})^8}{\epsilon^8(1-\gamma)^8}$$

$$=\Big\|\frac{1}{\mu}\Big\|_\infty^8 \frac{c_2|\mathcal{A}|^{18}(H+1)^{10}(R^*)^8}{(1-\gamma)^{16}\delta^{10}\epsilon^8}\Big[(1-\gamma^{H+1})^8 + \Big(\sum_{h=0}^{H-1}(((1-\gamma^{h+1})\gamma^{H-h-1})^{\frac{1}{2}})\Big)^8 \sum_{h=0}^{H-1}(1-\gamma^{h+1})^4(\gamma^{H-h-1})^4\Big]$$

$$\le\Big\|\frac{1}{\mu}\Big\|_\infty^8 \frac{c_2|\mathcal{A}|^{18}(H+1)^{10}(R^*)^8}{(1-\gamma)^{16}\delta^{10}\epsilon^8}\Big[(1-\gamma^{H+1})^8 + H^3\sum_{h=0}^{H-1}(((1-\gamma^{h+1})\gamma^{H-h-1})^4 \sum_{h=0}^{H-1}((1-\gamma^{h+1})\gamma^{H-h-1})^4\Big]$$

$$=\Big\|\frac{1}{\mu}\Big\|_\infty^8 \frac{c_2|\mathcal{A}|^{18}(H+1)^{10}(R^*)^8}{(1-\gamma)^{16}\delta^{10}\epsilon^8}\Big[(1-\gamma^{H+1})^8 + H^7\Big(\sum_{h=0}^{H-1}((1-\gamma^{h+1})\gamma^{H-h-1})^4\Big)^2\Big]$$

$$\le\Big\|\frac{1}{\mu}\Big\|_\infty^8 \frac{c_2|\mathcal{A}|^{18}(H+1)^{10}(R^*)^8}{(1-\gamma)^{16}\delta^{10}\epsilon^8}\Big[(1-\gamma^{H+1})^8 + H^8\sum_{h=0}^{H-1}(\gamma^{H-h-1}-\gamma^H)^8\Big],$$

where we used Jensen's inequality twice and defined the natural numbers $c_1 := 5*(144)^4$ and $c_2 := c_1*6^8$ which are independent of any model parameters. Next, since $\gamma \in (0,1)$, we have

$$\sum_{h=0}^{H-1}(\gamma^{H-h-1}-\gamma^H)^8 \le \sum_{h=0}^{H-1}(\gamma^{H-h-1})^8 \le \sum_{h=0}^{H-1}(\gamma^h)^8$$

$$\le \sum_{h=0}^{H-1}\gamma^h \le \frac{1}{1-\gamma}.$$

and obtain

$$\sum_{h=0}^{H} N_h K_h \le \Big\|\frac{1}{\mu}\Big\|_\infty^8 \frac{c_2|\mathcal{A}|^{18}(H+1)^{10}(R^*)^8}{(1-\gamma)^{16}\delta^{10}\epsilon^8}\Big[(1-\gamma^{H+1})^8 + H^8\sum_{h=0}^{H-1}(\gamma^{H-h-1}-\gamma^H)^8\Big]$$

$$\le \Big\|\frac{1}{\mu}\Big\|_\infty^8 \frac{c_2|\mathcal{A}|^{18}(H+1)^{10}(R^*)^8}{(1-\gamma)^{16}\delta^{10}\epsilon^8}\Big[1 + H^8\frac{1}{1-\gamma}\Big].$$

Finally, for $H = \lceil \log_\gamma\big(\frac{(1-\gamma)^2\epsilon}{6R^*}\big)\rceil = \Big\lceil\frac{\log((1-\gamma)^{-1}\epsilon^{-1}2R^*)}{\log(\gamma^{-1})}\Big\rceil$, there holds

$$\sum_{h=0}^{H} N_h K_h \le \Big\|\frac{1}{\mu}\Big\|_\infty^8 \frac{c_2|\mathcal{A}|^{18}(R^*)^8}{(1-\gamma)^{17}\delta^{10}\epsilon^8}\Big\lceil\frac{\log((1-\gamma)^{-1}\epsilon^{-1}2R^*)}{\log(\gamma^{-1})}\Big\rceil^{18}.$$

$\square$

