# OpenReview forum: "Structure Matters: Dynamic Policy Gradient"
_NeurIPS.cc/2025/Conference — NeurIPS 2025 poster_

### Official Review · Reviewer_6ZgJ · 2025-06-27

**Clarity:** 3
**Significance:** 4
**Originality:** 3
**Rating:** 5
**Confidence:** 3

**Summary:**

The authors consider policy gradient for $\gamma$-discounted infinite-horizon tabular reinforcement learning. They introduce Dynamic Policy Gradient (DynPG), a variant of policy gradient that exploits the Markovian property of the environment and thus reduces the worst-case error dependency on the effective horizon $(1-\gamma)^{-1}$ from exponential to polynomial (with exponent 4). Furthermore, they hint at better compatibility of their algorithm with actor-critic based methods due to independence of the value function from the current policy at future steps. Empirically, they show substantial improvements over vanilla policy gradient in a fabricated setting, though the additional memory cost of DynPG can quickly become prohibitively expensive in real-world implementations. The suggested actor-critic variant (DynAC) seems promising to alleviate the memory bottleneck, but lacks any theoretical or empirical analysis.

**Questions:**

> Firstly, we observe that DynPG effectively reduces the variance in gradient estimation through the utilization of non-changing future policies. In contrast, the estimation of the gradient in vanilla PG involves assessing Q-values based on the stationary policy $\pi_\theta$, which changes during training.

I thought this work would not consider any Q-value estimation, which would cross the line into actor-critic territory. So how exactly is the gradient variance reduced?

You argue that the total sample complexity is lower for DynPG compared to standard policy gradient. But if DynPG is run for longer than H steps, it becomes more expensive than vanilla policy gradient. Why does this not become an issue?

The value function is always written as an explicit function of the initial state distribution
$\mu$. It seems, however, that the optimal policy never depends on this state distribution, so why
is $\mu$ always explicitly included in all formulas?

> The tabular softmax policy can represent any deterministic and therefore optimal policy.

Can it not, more generally, represent any stationary policy?

Typos:
> In this section we provide estimates for exact gradient to keep notation simple

Do you mean “estimates”, or rather “bounds”?
> We formally quantify these errors in the following lemma

Did you rather mean “proposition”?

In Equation (5) there seems to be a typo where $V_0^{\hat \pi_0^*}$ should simply be $V_0$.

**Ethical Concerns:**

["NO or VERY MINOR ethics concerns only"]

**Final Justification:**

The authors were able to answer all my questions to a satisfactory degree. There do not remain any open points or points of disagreement, and my assessment that the work is a strong contribution has been reinforced.

**Limitations:**

yes

**Quality:**

3

**Strengths And Weaknesses:**

The paper is very well-written in all but section 4, where it is decently well-written. The contributions are effectively contextualized in prior literature. The proposed algorithm is elegant and shows substantial worst-case improvements on the effective horizon compared to standard approaches. It is also practical in the sense that linearizing the value $V$ as a function of the last step policy $\pi_h$ allows aggressive optimization.

However, DynPG is impractical in the sense that the memory cost becomes quickly prohibitively expensive. Also, the estimation of $\nabla_{\theta_n} V_{h+1}^{\{\pi_\theta, \Lambda\}}(\mu)$ is not sufficiently opened up. As a policy gradient algorithm one would assume complete rollouts + score trick, but at multiple points one hints at the use of Q-value functions, which rather indicates the use of DynAC.

---

> ### Author Rebuttal · Authors · 2025-07-29
>
> Dear reviewer,
>
> We sincerely thank you for the time spent evaluating our work and for the positive feedback.
>
> **Question 1 - variance reduction and Q-values:**
>
> You are correct that this work does not consider any Q-value estimation. Instead we sample one rollout - using the already trained (and therefore deterministic) policies - to estimate the gradient. Thus the variance in the gradient estimation of DynPG has less variance compared to vanilla PG. We agree that the formulation in this sentence is not accurate as plain vanilla PG also does not use Q-value estimation but rather uses a rollout under the current stationary policy to estimate the gradient. As this policy changes during the stochastic training, the variance is higher.
> We will reformulate this sentence accordingly to make this more precise.
>
> **Question 2 - runtime DynPG and complexity comparison DynPG vs vanilla PG:**
>
> Please note that the horizon $H$ in DynPG can be specified explicitly when comparing the complexity of DynPG and vanilla PG. Thus we can specify $H$, such that convergence of DynPG is guaranteed (see Theorem 4.8 and the detailed versions in the appendix for the specification of $ H$). This allows us to determine the upper bounds of DynPG in Theorem 4.8 only with respect to the stationary MDP parameters and more precisely independent of $H$.
>
> **Question 3 - dependence on $\mu$:**
> The choice of the initial state distribution $\mu$ is only crucial for training and the choice of $\mu$ influences the number of gradient steps until convergence in one epoch (see Theorem 4.6). Once the policy is converged and close to optimal (i.e. optimal in every state $s\in \mathcal{S}$), we can use it in the future trainings. The policy itself is then independent of $\mu$. Note that we need $\mu(s) >0 $ for every $s\in \mathcal{S}$ for this to hold.
>
>
> **Question 4 - class of softmax policies:**
>
> Yes you are right, softmax policies can approximate any stationary policy. With this sentence we just wanted to emphasize that softmax policies are suitable to approximate optimal policies. We will change this sentence to be: The tabular softmax policy can approximate any stationary and therefore also any optimal policy.
>
> **Typos:**
>
> Thank you for bringing this to our attention. We changed the two sentences accordingly and corrected the typo in Equation (5) for the camera ready version.

---

> > ### Comment · Reviewer_6ZgJ · 2025-08-01
> >
> > Thank you for addressing my concerns and open questions, as well as reaffirming the strength of the paper.

---

### Official Review · Reviewer_4fU7 · 2025-06-30

**Clarity:** 3
**Significance:** 3
**Originality:** 3
**Rating:** 5
**Confidence:** 3

**Summary:**

This paper introduces a new reinforcement learning algorithm called Dynamic Policy Gradient (DynPG) for solving $\gamma$-discounted infinite-horizon tabular MDPs. Unlike vanilla policy gradient (PG) methods that optimize a stationary policy over the full horizon, DynPG decomposes the problem into a sequence of contextual bandit problems. Each iteration optimizes only the first decision epoch while reusing previously trained policies for future steps, mimicking dynamic programming structure.

The authors prove that DynPG achieves global convergence under tabular softmax parametrization. Specifically, it avoids the exponential dependence on $(1 - \gamma)^{-1}$ found in vanilla PG and achieves a sample complexity of $\tilde{O}((1 - \gamma)^{-4} \varepsilon^{-1})$ to reach $\varepsilon$-optimality. Theoretical analysis includes an explicit error decomposition and optimization schedule. A synthetic experiment further demonstrates that DynPG mitigates the high variance and committal behavior that vanilla PG can suffer from.

**Questions:**

I have several questions about the future use of this algorithm and some notations.

1: If the MDP has infinite states such as linear MDP, does this setting lead to any additional challenges to your method and why?

2: If $\mu$ is a Dirac measure, what would the value of $||\frac{1}{\mu}||_{\infty}$ be? I think it should be 1 since you only need to consider starting from one single state, but please correct me if I'm wrong. I think it would be much better is there could be a remark about it.

**Ethical Concerns:**

["NO or VERY MINOR ethics concerns only"]

**Final Justification:**

The authors have answered my questions about the paper and I think this paper is well-motivated with solid theory. I will keep my score 5 about the paper.

**Limitations:**

yes.

**Paper Formatting Concerns:**

To my knowledge, there is no significant paper formatting concerns in this paper.

**Quality:**

3

**Strengths And Weaknesses:**

Strengths: This paper presents a conceptually elegant and practically meaningful idea: by incrementally training over increasing horizons, the global infinite-horizon reinforcement learning problem is decomposed into a sequence of simpler contextual bandit problems. This structural decomposition not only improves the tractability of policy optimization but also explicitly leverages the Markov property of the MDP. The proposed algorithm, DynPG, is concise and intuitive---each iteration requires only the specification of the sample size $N_h$ and gradient step schedule for the current horizon $h$, making it easy to implement.

The theoretical analysis is rigorous and well-organized.The authors also derive non-asymptotic convergence bounds under softmax parametrization with polynomial dependence on $(1-\gamma)^{-1}$, which contrasts with known exponential lower bounds for vanilla PG.
Overall, this work makes a solid theoretical contribution and opens up promising directions for designing structure-aware reinforcement learning algorithms.

Weaknesses: I think a potential weakness is about the experiment. The experiment of this paper is based on a single, synthetic tabular MDP. While this example is carefully constructed to illustrate the benefits of DynPG over vanilla PG, additional empirical results on more diverse environments (even small-scale function approximation settings) would help validate the practical utility of the approach.

---

> ### Author Rebuttal · Authors · 2025-07-29
>
> Dear reviewer,
>
> We sincerely thank you for the time spent evaluating our work and for the positive feedback.
>
> **Question 1 - infinite state space:**
>
> Only the convergence analysis is restricted to the tabular case. The method itself, however, can also be applied in non-tabular environments. Algorithm 1 in the paper is not depended on a specific MDP setting or policy parametrization. More specific, the algorithm can easily deal with infinite state spaces. One could use a linear softmax policy parametrization in linear MDPs, but any other parametrization will work as well. One can also use DynPG in problems with continuous action spaces.
>
> **Question 2 - Choice of $\mu$:**
>
> If $\mu$ is a Dirac measure, then $\lVert \frac{1}{\mu} \rVert_\infty = \infty$, as $\mu(s)=0$ for all but one state $s$. For the algorithm to work, we cannot choose $\mu$ to be a Dirac measure. The reason for this is straightforward. If $\mu$ is a Dirac measure, the algorithm learns only the behavior corresponding to that single state in each epoch and gains no insight into how it should act in other states. Therefore, for training purposes, it is essential that $\mu(s)>0$ for all $s\in\mathcal{S}$. Once training is complete, however, the initial state may be chosen deterministically. We will add a comment on this in the final version of the manuscript.

---

> > ### Comment · Reviewer_4fU7 · 2025-08-05
> >
> > I sincerely thank the authors for correcting me. I believe that my concern have already been resolved and I will maintain my acceptance score of this paper.

---

### Official Review · Reviewer_qBFH · 2025-07-01

**Clarity:** 3
**Significance:** 3
**Originality:** 3
**Rating:** 5
**Confidence:** 4

**Summary:**

The authors propose an algorithm that combines dynamic programming with policy gradient in reinforcement learning. Theoretical upper bounds for convergence rate of the proposed algorithm are derived and shown to avoid the potentially exponential dependency on the effective horizon $(1-\gamma)^{-1}$ in a vanilla policy gradient approach. This is numerically demonstrated on a toy example.

**Questions:**

In Algorithm 1, perhaps steps 4 and 6 can be made more explicit.

In Algorithm 1, step 3 initializes each new policy from scratch. One would imagine that the optimal policies between consecutive $h$ are similar and therefore initializing $\theta_0$ with the previous $\theta$ might have some benefits.

I would suggest another round of proof-reading to fix many of the missing words in the main text.

**Ethical Concerns:**

["NO or VERY MINOR ethics concerns only"]

**Final Justification:**

I maintain my acceptance recommendation.

**Limitations:**

yes

**Quality:**

4

**Strengths And Weaknesses:**

Strengths:

The main strength of the paper is the proposed algorithm together with the analysis of its convergence rate. It is convincing how the DP steps of the algorithm circumvent the limitation of a vanilla policy gradient approach, and this idea can perhaps be implemented and incorporated in existing, more practical RL algorithms. The presentation of the key results in the paper is generally clear. I did not check the proofs but results seem believable.

Weaknesses:

As usual in this kind of works, there is a question of the practical relevance of the proposed approach. The authors do state clearly that in many MDPs the vanilla policy gradient approach will do better. The issue of exploration as well as the challenge in estimating the gradient are omitted in the analysis. These issues have more practical relevance and may overshadow the gain from the proposed approach in practice.

---

> ### Author Rebuttal · Authors · 2025-07-29
>
> Dear reviewer,
>
> We sincerely thank you for the time spent evaluating our work and for the positive feedback.
>
> We would like to emphasize that we do not expect PG to outperform DynPG in most scenarios. Rather, we anticipate that in cases where PG performs well, DynPG will exhibit comparable performance, not worse performance. We will reformulate our formulation in line 226 to make this more precise. Additionally, please keep in mind that computing gradients in DynPG is actually simpler than in vanilla PG. The derivative with respect to the time-dependent policy has a simpler form (cf. Theorem C.3 with the gradient estimator used in PG). We will make this more explicit in the text.
>
> **Question 1 - Explanation of line 4 and 6 in the algorithm:**
>
> Line 6 of the algorithm specifies that $G$ should be an approximation, i.e. an estimator, of the exact gradient. In Theorem C.3, we express the true gradient as the expectation of log-probabilities multiplied by the future Q-function in a policy gradient theorem. This formulation enables an unbiased estimation of the gradient via straightforward Monte Carlo sampling. The estimation is simpler than in vanilla PG.
>
> Line 4 of the algorithm states that the hyperparameters $\eta_h$ and $N_h$ must be set. In Remark 4.4 we derive implications for how these parameters should be chosen based on the theoretical results.
> We will add remarks to clarify this in the final version of the manuscript.
>
> **Question 2 - Initialization of parameters for new epochs:**
>
> Copying the parameters from previous trainings as initialization for the new training epoch can indeed be a very good idea! It might be able to speed up the convergence in future epochs. Unfortunately, this is not always the case. Consider the example presented in the paper. Copying the weights from previous epochs would in this case lead to the same committal behavior effect as visible in vanilla PG. In general, we can therefore not guarantee improved convergence behavior when initializing based on previous trainings.
>
> **Question 3 - proof-reading:**
>
> Thank you for bringing this to our attention. We will thoroughly review the main text for any missing words before submitting the final version.

---

### Official Review · Reviewer_Q7Hh · 2025-07-01

**Clarity:** 4
**Significance:** 3
**Originality:** 3
**Rating:** 4
**Confidence:** 3

**Summary:**

This work develops DynPG: Dynamic Policy Gradient, an algorithm that exploits the structure of the MDP, combining dynamic programming with vanilla policy gradient. The authors proceed to study the newly proposed method in the $\gamma$-discounted, infinite horizon tabular MDPs, using softmax policies and establish finite sample convergence to the stationary optimal policy. Exploiting the MDP structure enables the algorithm to scale polynomially in the effective horizon and circumvents a fundamental limitation of vanilla policy gradient.

**Questions:**

Answering these questions and including the discussions will definitely make the paper stronger:

- PG methods and AC variants, especially with the softmax policy are mostly used in non tabular MDP settings. What can we expect in these scenarios? Can we extend the analysis to this setting? and how does the DynPG method apply here?
- The example in Figure 2 compares DynPG to vanilla PG (with REINFORCE, no baseline). What happens when the baseline exploits the MDP structure (say use a Q or an advantage function), does it still fail in the example? Do we still have the same exponential dependence on the effective horizon?
- Scaling DynPG to larger problems require an AC variant. Its theoretical analysis is out of scope, but I would have liked comparing it to classical Actor Critic. What are the main algorithmic differences? What should we intuitively expect from DynAC?

**Ethical Concerns:**

["NO or VERY MINOR ethics concerns only"]

**Final Justification:**

My questions were addressed satisfactorily. While the contribution is primarily theoretical, it is both important and relevant to the community. I would have appreciated a deeper discussion of the practical implementation details and implications of the method.

Overall, I support the acceptance of this work.

**Limitations:**

Yes. The limitations of the analysis and the proposed algorithm were covered mostly.

**Quality:**

3

**Strengths And Weaknesses:**

I like the paper overall, and think that its strengths overweight its weaknesses. These are listed below:

**Strengths:**

- The paper is written in a clear and accessible way.
- The theoretical contribution is relevant to the community and answers an important question.
- The paper acknowledges a big part of the limitation of the proposed method and suggests improvements.


**Weaknesses:**
- The analysis require somewhat strong assumptions (access to exact gradients for example).
- It is a theoretical paper, but I would have loved to see more experimental discussions.
- There is a lack of discussion and some questions are left unanswered. This is developed more in the questions section.

---

> ### Author Rebuttal · Authors · 2025-07-29
>
> Dear reviewer,
>
> We sincerely thank you for the time spent evaluating our work and for the positive feedback.
>
>
> First, we would like to point out that we relax the assumption of exact gradients in Appendix F and instead conduct our analysis under stochastic, sampled gradients. Even in this setting, we observe an improved convergence rate of DynPG compared to regularized vanilla PG (cf. Remark F.7).
>
> In the following, we address your questions regarding the practical applicability of DynPG and its performance. Additional experiments are planned for future research.
>
> **Question 1 – DynPG in Non-Tabular Settings:**
>
> Only the convergence analysis is restricted to the tabular case. The method itself, however, can also be applied in non-tabular environments. Algorithm 1 in the paper is not depended on a specific MDP setting or policy parametrization.
> At present, the analysis using softmax cannot be extended to infinite state spaces. In such settings, it is open to establish a convergence rate based on properties of the loss landscape such as dominated gradient property, which is crucial for proving convergence in our setting. Nevertheless, in scenarios where standard algorithms suffer from committal behavior, we still expect DynPG to exhibit superior performance.
>
> **Question 2 - DynPG vs. PG with baseline in our Example:**
>
> The possible exponential dependence of the effective horizon has been proved in the setting of exact gradients. Using baselines does not change the exact gradients at all, so the counter examples stay untouched using a baseline. Nonetheless, for the stochastic setting your question is very good and to our knowledge there is no answer. We performed additional experiments on our toy example with baselines, in particular using the estimated value function of the current policy. The baseline did not improve a lot, on the example DynPG was still significantly better.
>
> **Question 3 - DynAC vs. AC:**
>
> The difference of DynAC and AC is similar to DynPG and PG. We stated DynAC in Section E. In DynAC we use an actor critic approach to learn a new policy for each epoch (as it is done in DynPG using standard PG). Compared to that in AC we learn one stationary policy applied in every epoch. Intuitively, the situation should be similar to our toy examples, delicate environments with dead ends should be favorable to DynAC while (as always) the simpler algorithms will perform equally well on simple environments.

---

> > ### Comment · Reviewer_Q7Hh · 2025-08-01
> >
> > Thank you for addressing my questions. While I would have appreciated a more in-depth discussion on the practical applications of the approach, the theoretical contribution of the paper is already solid and relevant to the community.

---

### Decision · Program_Chairs · 2025-09-17

**Decision:**

Accept (poster)

**Comment:**

The authors consider RL problems formulated as discounted infinite-horizon tabular MDPs, and introduce a framework called dynamic policy gradient (DynPG) that decomposes the problem into a sequence of contextual bandit problems, and combines some of the advantages of both a dynamic programming perspective and a policy gradient perspective.  They provide theoretical bounds for the convergence rate that are much better than the worst-case (exponential in the effective horizon) bounds for vanilla policy gradient methods.

The rebuttal period discussion for this paper mostly focused on technical points; the paper was well written, and the reviewers seem to have mostly understood the authors without additional discussion.  Everyone came into the rebuttal period fairly positive and remained so afterward.

The paper is primarily theoretical, as the authors explicitly acknowledge in the initial draft and as the reviewers also observed.  However, the reviewers were positive about the theoretical contributions ("convincing", "relevant to the community and answers an important question", "conceptually elegant and practically meaningful").  Though the algorithm proposed may not always be the right algorithm on its own, the reviewers commented that it seems plausible that the ideas can be implemented in existing, perhaps more practical RL algorithms.  This seems like it would garner quite a bit of community interest and inspire further work.